# Semantic-Enhanced Time-Series Forecasting via Large Language Models

**Hao Liu[1], Xiaoxing Zhang[2], Chun Yang[1], Xiaobin Zhu[1]** *
[1]University of Science and Technology Beijing, [2]Yizhi, China Telecom
d202410441@xs.ustb.edu.cn, zhangxx7@chinatelecom.cn,
chunyang@ustb.edu.cn, zhuxiaobin@ustb.edu.cn

## Abstract

Time series forecasting plays a significant role in finance, energy, meteorology, and IoT applications. Recent studies have leveraged the generalization capabilities of large language models (LLMs) to adapt to time series forecasting, achieving promising performance. However, existing studies focus on token-level modal alignment, instead of bridging the intrinsic modality gap between linguistic knowledge structures and time series data patterns, greatly limiting the semantic representation. To address this issue, we propose a novel Semantic-Enhanced LLM (SE-LLM) that explores the inherent periodicity and anomalous characteristics of time series to embed into the semantic space to enhance the token embedding. This process enhances the interpretability of tokens for LLMs, thereby activating the potential of LLMs for temporal sequence analysis. Moreover, existing Transformer-based LLMs excel at capturing long-range dependencies but are weak at modeling short-term anomalies in time-series data. Hence, we propose a plugin module embedded within self-attention that models long-term and short-term dependencies to effectively adapt LLMs to time-series analysis. Our approach freezes the LLM and reduces the sequence dimensionality of tokens, greatly reducing computational consumption. Experiments demonstrate the superiority performance of our SE-LLM against the state-of-the-art (SOTA) methods.

## 1 Introduction

Time series forecasting has widespread applications in various tasks, such as finance, energy, meteorology, and industrial IoT (Liu et al., 2016; Arslan, 2022; Sherstinsky, 2018). Recently, Transformer-based LLMs (Radford et al., 2019; Devlin et al., 2019; Yang et al., 2024a; Touvron et al., 2023) are prevalent in time series forecasting (Jin et al., 2024; Liu et al., 2024c; 2025; Hu et al., 2025) by integrating multi-domain knowledge to model temporal dependencies, thereby enhancing time series analysis effectively.

Time series data fundamentally differs from linguistic structures in nature. Existing methods often convert temporal information into textual prompts to activate the potential of LLMs. GPT4mTS (Jia et al., 2024a) introduces a prompt-based framework that jointly processes temporal data and textual information. Zhou et al. (Zhou et al., 2023) processed time-series data by randomly initializing the embedding layer and directly performing serialized analysis through frozen LLMs. However, the inherent modality gap between temporal and linguistic data resulted in low-quality embeddings. In this regard, LLM4TS (Chang et al., 2025) introduces a unified framework that fuses token, positional, and temporal embeddings to maintain modality consistency. Time-LLM (Jin et al., 2024) converts time-series data into text prototype representations and integrates textual prompts to enhance time series analysis. Meanwhile, S2IP-LLM (Pan et al., 2024a) incorporates time-series data into LLM's semantic space as prompts, compensating for the lack of textual prompts. However, incorporating textual information may introduce noise (Liu et al., 2025), and the text descriptions incur additional computational cost (Lin et al., 2024; Ma et al., 2023; Shang et al., 2024).

Most existing LLM-based methods mainly focus on token-level alignment (Hu et al., 2025), as shown in Fig. 1 (a) (Pan et al., 2024a; Liu et al., 2025). The joint space serves as an implicit prompt,

---

*Corresponding author. The code is available at https://github.com/LH325/SE-LLM.

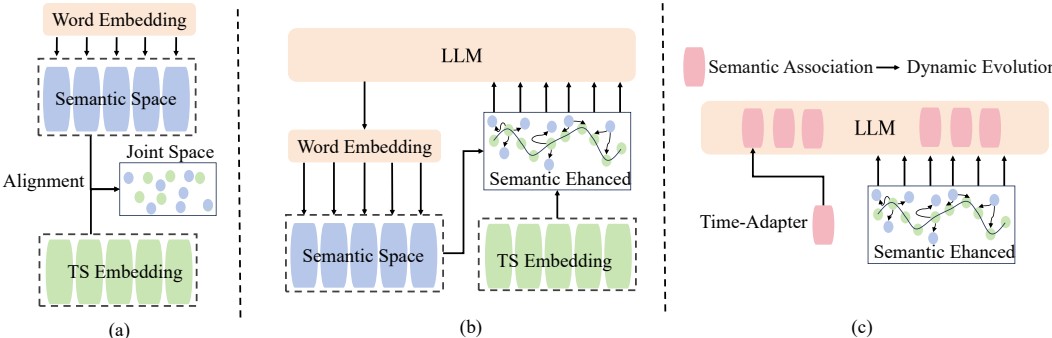

Figure 1: (a) The input time series is mapped to obtain time series (TS) Embeddings. Next, the feature space is aligned with the semantic space derived from the pre-trained word token embedding. (b) Temporal patterns are injected into the joint space and use implicit prompts to LLM for forecasting. (c) Embedding Time Adapter enhances learning of temporal patterns.

guiding the LLM in conducting time series forecasting. Semantic tokenization eliminates the dependency on auxiliary textual descriptions. However, this token-level alignment overlooks temporal and channel dependencies, making it difficult to capture dynamic temporal patterns. To address these limitations, our analysis identifies a critical oversight in existing LLM-based approaches (Jin et al., 2024; Liu et al., 2024c; Chang et al., 2025; Zhou et al., 2023). Regarding the semantic space composed of tokens and time steps, we propose the Temporal-Semantic Cross-Correlation (TSCC) Module to obtain enhanced semantics that represent temporal patterns, emphasize anomaly and de-anomaly patterns, thereby enhancing the interpretability of the token embeddings, as shown in Fig. 1 (b). Although we provide semantic guidance with high interpretability from LLMs, there exists an inherent difference between LLMs' understanding of linguistic knowledge and temporal pattern. We propose to freeze the LLM and introduce an adapter to transform the model's semantic information into the capability of modeling dynamic changes in time series, as shown in Fig. 1 (c).

In summary, we propose Semantic-Enhanced LLM, a novel framework that enhances LLM for time series forecasting with two key components: the Temporal-Semantic Cross-Correlation (TSCC) module and the Time-Adapter architecture. The TSCC captures cross-correlation of temporal embedding and semantic space with temporal dependencies and channel-wise relationships, which are embedded into token embeddings to adapt LLM to temporal features. The Time-Adapter enhances forecasting capability by overcoming the limitation of LLMs in modeling both short-term and long-term dependencies based on historical time segments. This enables pretrained LLMs to forecast time series effectively. Our main contributions are four-fold:

- We introduce a novel SE-LLM to bridge the inherent modal differences between time series and language data, activating the generalization capability of LLMs for time-series analysis.
- We propose a novel Temporal-Semantic Cross-Correlation (TSCC) module to enhance token embedding by endowing semantic information with temporal patterns, which improves temporal analysis.
- We propose a Time-Adapter to bridge the modality gap between LLMs and time-series data, effectively modeling both long-term dependencies and short-term patterns.
- Extensive experiments verify the superior performance and efficiency of our SE-LLM.

## 2 RELATED WORKS

### 2.1 TIME SERIES FORECASTING

Traditional time series forecasting methods, such as ARIMA (Liu et al., 2016) and Prophet (Arslan, 2022), often relied on statistical and classical machine learning techniques. The rise of deep learning brought Recurrent Neural Networks (RNNs) (Sherstinsky, 2018; Qin et al., 2017; Yeo et al., 2018; Khaldi et al., 2023), which excelled at modeling temporal dependencies. Long Short-Term

Memory (LSTM) networks(Kong et al., 2025; Shahzad et al., 2023) improved RNNs by addressing vanishing gradients and long-range dependencies. Advanced methods such as PGN (Jia et al., 2024b) further enhance long-term temporal relationship modeling, and improved computational efficiency. Concurrently, Graph Neural Networks (GNNs) (Cao et al., 2020; Pan et al., 2024b; Deng et al., 2020) excelled at capturing spatial and structural dependencies in spatiotemporal contexts. Recently, Transformer-based architectures (Chen et al., 2021; Zhang & Yan, 2023; Wang et al., 2024) have dominated due to their ability to capture global temporal patterns. However, these models struggle with generalizability and flexibility in highly dynamic, real-world time series analysis.

## 2.2 LLM-BASED TIME SERIES FORECASTING

Recently, LLMs (Radford et al., 2019; Touvron et al., 2023; Zhang et al., 2022; DeepSeek-AI, 2024; Yang et al., 2024a) have developed rapidly, and demonstrated significant potential in advancing time-series forecasting. Gruver *et al.* (Gruver et al., 2023) and Xue *et al.* (Xue & Salim, 2023) reformulated forecasting as a sentence-to-sentence translation task to bridge the modality gap. Time-LLM (Jin et al., 2024) leverages token-level embedding as a prompt to enhance the textual description to enhance LLM's understanding of temporal concepts. Pan *et al.* (Pan et al., 2024a) integrate semantic space with embedding as a prompt to guide LLM. Liu *et al.* (Liu et al., 2025) indicate that the embedding of textual information is weak, and they proposed an entangled reliable embedding. Hu *et al.* (Hu et al., 2025) introduced FSCA, which utilizes Graph Neural Networks (GNNs) to ensure structural and logical alignment. AutoTimes (Liu et al., 2024c) explored autoregressive forecasting with timestamp embedding to improve LLM's performance. Notably, the aforementioned studies and most research (Jia et al., 2024a) employ a frozen LLM approach for time series learning, indicating that LLMs have already acquired rich general semantic recognition capabilities during the pre-training weight. If all parameters are directly fine-tuned, these general capabilities may be compromised, leading to unstable performance in time series tasks. Current research merely leverages the generalization ability of LLMs for downstream tasks, suggesting that the inherent architecture of LLMs is not well-suited for time series forecasting.

## 2.3 ADAPTER-BASED LLM FINE-TUNE

Existing methods show that fine-tuning LLMs (Jin et al., 2024; Zhou et al., 2023; Gruver et al., 2023) not only consumes computational resources but also degrades forecasting performance across data from different domains. Adapter-based fine-tuning techniques (Hu et al., 2022; Guo et al., 2024; Zhang et al., 2023; Lester et al., 2021; Zhang et al., 2025) have demonstrated efficient adaptation to task-specific distributions in various tasks (Liu et al., 2024a; Chen et al., 2024; Mu et al., 2025). Empirical studies, e.g., Gupta *et al.*(Gupta et al., 2024), validate the effectiveness of LoRA-style fine-tuning in boosting time-series forecasting performance. AutoTimes(Liu et al., 2024c) incorporates LoRA (Hu et al., 2022) into GPT-2 (Radford et al., 2019), achieving substantial improvements in long-range prediction scenarios. However, the success of these fine-tuning techniques is highly contingent upon algorithmic design choices (Pan et al., 2024a; Zhou et al., 2023; Jin et al., 2024), and the transferability of these methods across diverse time-series domains remains unclear. We contend that current adapters, while enhancing LLMs' understanding of semantic structures, are not designed to capture temporal patterns effectively. In contrast, the Time-Adapter is specifically designed to model short- and long-term dependencies, helping LLMs adapt to temporal forecasting tasks.

# 3 METHODS

## 3.1 OVERVIEW OF SE-LLM

The proposed SE-LLM model consists of two primary components: the TSCC and the Time-Adapter, as depicted in Fig. 2. The process begins with the input sequence, a time series data matrix with batch size $\mathbf{B}$ and sequence length $\mathbf{L}$.A sliding window operation is applied to partition the temporal dimension into smaller segments, transforming the original sequence $\mathbf{T}$ into a segmented representation $\tilde{\mathbf{T}} \in \mathbb{R}^{B \times N \times K}$, where $N$ denotes the number of segments and $K$ denotes the segment length. This operation reduces the effective sequence length processed by the LLM from $L$ to $N$, thereby reducing the self-attention complexity from $\mathcal{O}(L^2)$ to $\mathcal{O}(N^2)$ with respect to the num-

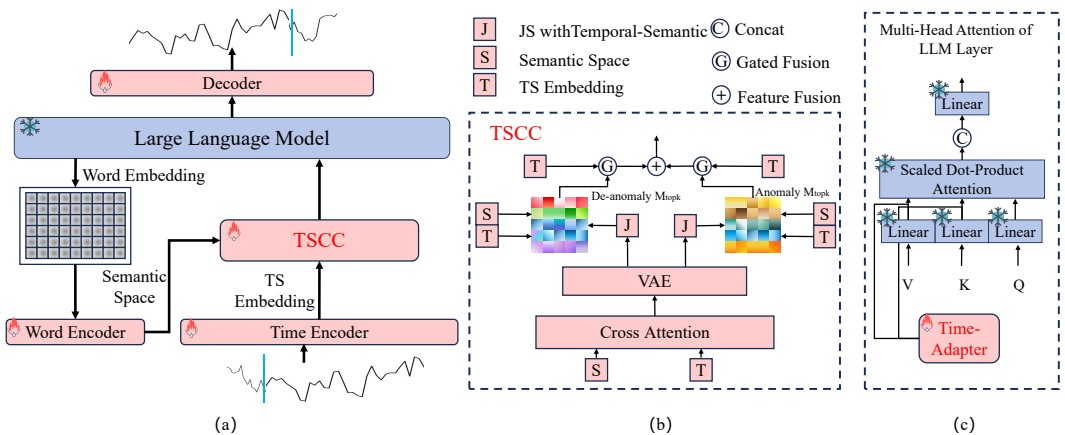

Figure 2: (a) An overview of SE-LLM. Red indicates trainable, while blue indicates frozen. (b) The overall framework of TSCC, with detailed processes referring to Fig. 3. (c) For the Transformer-based LLM, the Time-Adapter is embedded within the key and value vectors of the multi-head attention mechanism. JS is an abbreviation for Joint Space.

ber of temporal tokens. This design reduces self-attention and feed-forward network complexities, improving the model's efficiency.

**TS Embedding.** The temporal representation is passed through the Time Encoder module, which projects the data into a high-dimensional TS Embeddings, which can be formulated as:

$$\mathbf{H} = \mathcal{F}_2 \left( \sigma \left( \mathcal{F}_1(\tilde{\mathbf{T}}) \right) \right),\tag{1}$$

where $\mathbf{H} \in \mathbb{R}^{B \times N \times C}, \mathcal{F}_1, \mathcal{F}_2$ are linear layers, $\sigma$ is activation. This projection captures more expressive features of the temporal patterns within the data.

**Semantic Space.** We construct the semantic space by projecting the word embedding matrix of the pre-trained LLM through a linear layer. Specifically, the word embedding matrix $\mathbf{W} \in \mathbb{R}^{V \times C}$ is mapped to $\mathbf{S} \in \mathbb{R}^{K_s \times C}$ (Fang et al., 2025). The resulting semantic representations serve as general linguistic priors and are aligned with the TS embeddings for temporal modeling.

We input both the semantic space and temporal features into the TSCC to obtain enhanced semantics, as shown in Fig. 2(b). Following this, the LLM embeds Time-Adapter into the Multi-Head Attention to further refine the model's ability to understand temporal patterns in the time series data, as shown in Fig. 2(c). This component enhances the overall temporal pattern recognition by integrating additional contextual information, enabling better forecasting performance. Finally, the Decoder component decodes the output from the LLM to produce the final prediction as:

$$\mathbf{O} = \mathcal{F}_2 \left( \sigma \left( \mathcal{F}_1(\mathbf{Y}) \right) \right),\tag{2}$$

where $\mathbf{Y}$ is the LLM's output. The test processes are provided in Appendix A.

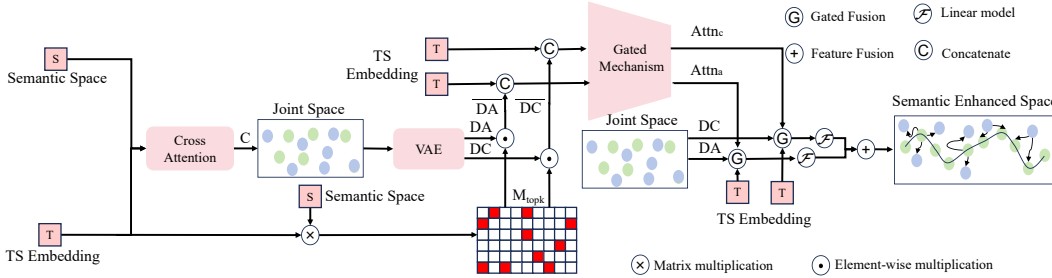

Figure 3: Architecture of TSCC module, where $\overline{\mathrm{DA}}$ and $\overline{\mathrm{DC}}$ correspond to "de-anomaly semantic" and "anomaly semantic".

## 3.2 TEMPORAL-SEMANTIC CROSS-CORRELATION

Fig. 3 is the architecture of the TSCC module. The core of this module is to infuse temporal patterns into the joint space. Firstly, we align the temporal embeddings with the semantic space through a cross-attention mechanism (Jin et al., 2024; Liu et al., 2025; Chang et al., 2025).

**Cross-Modality Alignment.** The Cross-Domain Attention Mechanism is capable of dynamically capturing complex associations between different modalities, and achieving efficient fusion through adaptive weight allocation. Hence, we employ Cross Attention to align temporal embeddings with the semantic space, thereby obtaining Joint Space (JS) $\mathbf{C} \in \mathbb{R}^{B \times N \times C}$. This can be formulated as:

$$\mathbf{C} = \mathbf{CrossAttn}(H, S). \tag{3}$$

**Anomaly Pattern Modeling.** In time series analysis, the anomaly noise in non-stationarity information lead to inaccurate model parameter estimation and increased prediction bias. We employ an Anomaly Modeling Variational Autoencoder (AM-VAE) (Cemgil et al., 2020) to estimate a latent distribution of the joint-space representation and reconstruct a semantic component from the sampled latent variable. Given the cross-attention output $\mathbf{C}$, the encoder predicts the latent mean and log-variance, from which a latent variable is sampled via the reparameterization trick.

---

**Algorithm 1** Anomaly Modeling via AM-VAE Module

---

**Require:** Cross-attention output: Joint Space $\mathbf{C} \in \mathbb{R}^{B \times N \times C}$
**Ensure:** Decomposed anomaly semantic $\mathbf{DC}$ and de-anomalized semantic $\mathbf{DA}$ with temporal pattern
  1: **Latent Feature Projection:** $\mathbf{V} \leftarrow \sigma\left(\mathcal{F}_e(\mathbf{C})\right)$, where $\mathcal{F}_e$: Linear mapping from $\mathbb{R}^C \rightarrow \mathbb{R}^n$
  2: **Latent Distribution Estimation:** $\boldsymbol{\mu}, \log \boldsymbol{\sigma}^2 \leftarrow \mathcal{F}_\mu(\mathbf{V}), \mathcal{F}_\sigma(\mathbf{V})$, where $\mathcal{F}_\mu, \mathcal{F}_\sigma$: Linear mapping from $\mathbb{R}^n \rightarrow \mathbb{R}^m$.
  3: **Reparameterization Trick:** Sample from latent space, $\mathbf{z} \leftarrow \boldsymbol{\mu} + \boldsymbol{\epsilon} \odot \exp\left(0.5 \cdot \log \boldsymbol{\sigma}^2\right) \leftrightarrow z = \mu + \epsilon \odot \sigma, \quad \boldsymbol{\epsilon} \sim \mathcal{N}(0, \mathbf{I})$.
  4: **Anomaly Sample**: Instead of directly modeling future observations, AM-VAE estimates a latent distribution of the joint temporal-semantic representation $\mathbf{C}$. The reparameterization trick enables stochastic sampling in the latent semantic space, allowing the decoder to reconstruct the anomaly-related semantic component.
  5: **Anomaly Semantic Modeling:** Decode latent variable to recover anomaly semantic, $\mathbf{DC} \leftarrow \mathcal{F}_d(\mathbf{z}), \quad \mathcal{F}_d : \mathbb{R}^m \rightarrow \mathbb{R}^C$
  6: **Anomaly Decomposition:** $\mathbf{DA} = \mathbf{C} - \mathbf{DC}, \quad \mathbf{DC} \in \mathbb{R}^{B \times N \times C}, \mathbf{DA} \in \mathbb{R}^{B \times N \times C}$, where $\mathbf{DC}$ represents the reconstructed anomaly-related semantic component in the joint space, while $\mathbf{DA}$ represents the joint space where anomalous semantics are removed.

---

**Structural Prior Infusion.** The temporal-semantic similarity matrix $\mathbf{M}$ is computed after applying L2 normalization to both temporal and semantic features, which reduces the influence of feature magnitude and makes the correlation score closer to cosine similarity:

$$\mathbf{M} = \mathrm{Norm}_2(\mathrm{Mean}_N(\mathbf{H})) \times \mathrm{Norm}_2(\mathbf{S})^T. \tag{4}$$

We average the temporal embeddings along the segment dimension and compute their L2-normalized similarity with semantic prototypes. The top-$K$ semantic prototypes are then selected according to this sample-level similarity and aggregated as a structural semantic prior:

$$\overline{\mathbf{DA}} = \mathrm{DA} \cdot \frac{1}{n} \sum_{i=1}^{n} \mathcal{P}(\mathrm{S_i}, \mathrm{m_{top_k}^i}), \quad \overline{\mathbf{DC}} = \mathrm{DC} \cdot \frac{1}{n} \sum_{i=1}^{n} \mathcal{P}(\mathrm{S_i}, \mathrm{m_{top_k}^i}). \tag{5}$$

The aggregated semantic prior is used to condition the subsequent gated fusion of the AM-VAE outputs, including the de-anomalized semantic representation $\mathbf{DA}$ and the anomaly-related semantic representation $\mathbf{DC}$.

For different datasets or model architectures, directly incorporating top-$K$ semantic cues may introduce additional noise or amplitude perturbations. To mitigate this effect, we normalize the aggregated top-$K$ semantic cues before using them for gate conditioning. It is worth noting that, when the softmax operation is applied along a singleton dimension after aggregation, it degenerates to an

all-one scaling factor and therefore should not be interpreted as an additional adaptive weighting mechanism. In this case, the selected top-$K$ semantic cues mainly serve as a structural prior for the gated fusion process rather than introducing extra learnable or adaptive weights.

**Channel Dependency Enhancement.** Although the enhanced joint semantics integrate information from TS embeddings, they may not fully preserve temporal patterns after cross-modal fusion. To mitigate this issue, we further model channel dependencies based on the temporally enhanced semantic representations. Specifically, we concatenate the TS embeddings $\mathbf{H}$ with $\overline{\mathbf{DA}}$ and $\overline{\mathbf{DC}}$ to enable channel-wise information interaction. An MLP is then employed to extract channel attention and enhance the semantic representations along the channel dimension:

$$\mathbf{Attn_a} = \mathbf{MLP}([\mathrm{H}, \overline{\mathrm{DA}}]), \tag{6}$$

where $\overline{\mathbf{DC}}$ is processed in the same way to get $\mathbf{Attn_c}$, and we will mainly focus on introducing $\mathbf{DA}$. The $\mathbf{Attn}$ module generates channel-wise gates conditioned on TS embeddings, making the semantic fusion sensitive to temporal representations. A gating mechanism is then used to fuse the joint space with the TS embeddings, injecting temporal patterns into the resulting joint space:

$$\mathbf{GA} = \mathcal{F}_{llm}(\mathrm{Attn_a} \odot \mathrm{H} + (1 - \mathrm{Attn_a}) \odot \mathrm{DA}), \tag{7}$$

where, $\mathcal{F}_{llm}$ is a linear layer that maps the enhanced semantics to the token space of the LLM. The resulting embeddings are enriched with semantic information that reflects temporal patterns. Finally, the de-anomalization ($\mathbf{GA} \in \mathbb{R}^{B \times N \times C}$) and anomaly ($\mathbf{GC} \in \mathbb{R}^{B \times N \times C}$) representations are mapped through a linear layer for alignment and fused to generate a unified semantic representation, which is more interpretable and suitable for LLMs analysis. It can be formulated as:

$$\mathbf{Y} = \mathbf{LLM}(\mathrm{GA} + \mathrm{GC}). \tag{8}$$

This represents the fusion of the enhanced de-anomalized ($\mathbf{GA}$) and anomalous ($\mathbf{GC}$) semantics enriched with temporal patterns. The resulting unified representation is then processed through the LLM to analyze these enhanced semantics and generate a more accurate prediction.

Figure 4: The architecture of Time-Adapter, a modular enhancement designed to improve the temporal modeling capabilities of LLMs. It comprises two linear layers and two LSTM modules, enabling effective capture of both long-term and short-term temporal dependencies.

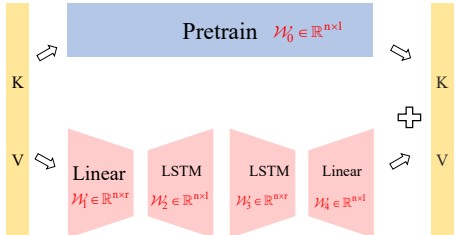

### 3.3 TIME-ADAPTER

Time-Adapter explicitly captures long-term and short-term dependencies, compensating for the Transformer's inherent weaknesses in temporal pattern extraction, as shown in Fig. 4. Specifically, building upon LoRA (Hu et al., 2022), we replace the low-rank matrix with dual linear layers and insert sequential LSTM (Hochreiter & Schmidhuber, 1997) units between them. These paths process temporal dependencies by leveraging long-range information from the Transformer's architecture through nonlinear transformations, thereby equipping the model with temporal data-handling capabilities during training. The Time-Adapter module explicitly captures long-term and short-term dependencies by processing the temporal patterns through LSTM paths. These dependencies are integrated into the key and value matrices of the multi-head attention mechanism. This enhancement enables the model to handle complex temporal patterns more effectively.

Our method consists of four key operations executed sequentially. First, a **Low-rank Projection** is applied, where a linear layer learns a low-rank matrix to reduce the input dimensionality. Then the **Long-term Dependency Modeling**, in which an LSTM (Zhang et al., 2022) maps the compressed features to a high-dimensional space, effectively capturing long-term temporal patterns. Finally, **Short-term Dependency Modeling** is performed, where a second LSTM processes the high-dimensional representation via a reverse projection (high-to-low mapping), isolating localized short-term dynamics. Finally, a linear layer integrates these refined temporal dependencies into the key ($\mathrm{k} \in \mathbb{R}^{B \times N \times C}$) and value ($\mathrm{v} \in \mathbb{R}^{B \times N \times C}$) matrices of the self-attention mechanism, enhancing their capacity to model temporal relationships.

# 4 EXPERIMENTS

The evaluation metrics, datasets, and baseline methods are provided in the Appendix B.1.

## 4.1 COMPARATIVE RESULTS

**Long-Term Forecasting.** The long-term forecasting results are provided in Table 1. SE-LLM achieves the best or highly competitive performance across most datasets. On the Traffic dataset, which contains highly nonlinear patterns with strong variability, our approach obtains a 4.7% reduction in MAE compared with the strongest baseline. On ECL, SE-LLM also achieves the best average performance and demonstrates strong stability under periodic consumption patterns. For ETTh1 and Solar, where seasonal and trend components are pronounced, our method maintains competitive accuracy across different forecasting horizons. Some results are unavailable because several previous methods do not provide accessible code or cannot be applied to specific datasets due to design constraints. All reported comparisons follow the same input length, forecasting horizons, and evaluation metrics. Overall, the results indicate that SE-LLM alleviates long-horizon degradation and preserves high accuracy across diverse data distributions, confirming its effectiveness in long-term forecasting.

Table 1: The input length of the SE-LLM is 672. The forecast horizon is $\{96, 192, 336, 720\}$. The best results are in **red** and the second best are **blue**. Full results are provided in Table 9.

| Models | SE-LLM | | TimeMixer++ 2025 | | TimeMOE 2025 | | Time-CMA 2025 | | TQNet 2025 | | LLM-TS 2025 | | AutoTimes-GPT2 2024 | | Time-LLM 2024 | | iTransFormer 2024 | |
|--------|-----|-----|-----|-----|-----|-----|-----|-----|-----|-----|-----|-----|-----|-----|-----|-----|-----|-----|
| Metrics | MSE | MAE | MSE | MAE | MSE | MAE | MSE | MAE | MSE | MAE | MSE | MAE | MSE | MAE | MSE | MAE | MSE | MAE |
| ETTh1 | 0.381 | 0.415 | 0.429 | 0.443 | 0.412 | 0.426 | 0.396 | 0.419 | 0.438 | 0.450 | 0.454 | 0.451 | 0.397 | 0.425 | 0.409 | 0.432 | 0.421 | 0.445 |
| Weather | 0.229 | 0.267 | 0.228 | 0.266 | 0.256 | 0.289 | 0.250 | 0.276 | 0.227 | 0.268 | 0.257 | 0.285 | 0.242 | 0.281 | 0.227 | 0.266 | 0.266 | 0.291 |
| Traffic | 0.386 | 0.261 | 0.404 | 0.311 | —— | —— | —— | —— | 0.400 | 0.278 | 0.618 | 0.333 | 0.406 | 0.276 | 0.402 | 0.294 | 0.384 | 0.274 |
| ECL | 0.161 | 0.255 | 0.163 | 0.257 | —— | —— | 0.174 | 0.269 | 0.161 | 0.259 | 0.173 | 0.266 | 0.173 | 0.268 | 0.170 | 0.275 | 0.164 | 0.258 |
| Solar | 0.192 | 0.242 | 0.194 | 0.256 | —— | —— | 0.227 | 0.276 | 0.230 | 0.270 | —— | —— | 0.207 | 0.246 | 0.234 | 0.293 | 0.213 | 0.291 |

**Short-Term Forecasting.** It refers to predicting target variables within a relatively brief future time horizon. We have meticulously selected algorithms suitable for short-term forecasting for comparison. SE-LLM was compared with the SOTA models in short-term forecasting on the M4 dataset, as shown in Table 2, and the results demonstrated that we achieved the best performance. Compared with the second-best results, SE-LLM reduces SMAPE, MASE, and OWA by approximately 0.42%, 0.13%, and 0.35%, respectively. This indicates that SE-LLM has the ability to capture short-term time-varying patterns through enhanced short-term dependencies.

Table 2: The M4 dataset includes yearly, quarterly, monthly, weekly, daily, and hourly data, the short-term forecasts averaged across all M4 subsets. Full results are provided in Table 10.

| Models | | SE-LLM | FSCA 2025 | Time-VLM 2025 | AutoTimes 2024 | S2IP-LLM 2024 | Time-LLM 2024 | FPT 2024 | iTransformer 2024 |
|--------|------|--------|------|------|------|------|------|------|------|
| Average | SMAPE | 11.778 | 11.828 | 11.894 | 11.831 | 12.021 | 11.983 | 11.991 | 12.142 |
| | MASE | 1.578 | 1.580 | 1.592 | 1.585 | 1.612 | 1.595 | 1.600 | 1.631 |
| | OWA | 0.847 | 0.850 | 0.855 | 0.850 | 0.857 | 0.859 | 0.861 | 0.874 |

**Zero-Shot forecasting.** Furthermore, the proposed SE-LLM exhibits generalization capability to unseen temporal patterns, indicating its effectiveness in zero-shot forecasting scenarios. The ability to generalize across different datasets is critical for this task, as the model must adapt to varying data distributions and temporal structures without prior training on those specific patterns. We conducted zero-shot prediction experiments on the univariate M3 and M4 datasets, which exhibit diverse temporal variation patterns and distinct data distributions. Unlike many existing methods that struggle with such generalization, our model leverages a learned latent representation to capture underlying temporal structures, thereby facilitating transferability across domains. Specifically, the M3 and M4 datasets present heterogeneous temporal patterns (e.g., ranging from quarterly to monthly or daily frequencies), and our approach benefits from strong generalization capability to model realistic temporal dynamics, even in the absence of direct training data for those patterns.

Table 3: The experimental results are based on the generalization of consistent temporal patterns. For non-matching patterns, we evaluate frequency transfers from Monthly to Weekly/Daily/Hourly in the M3 → M4 dataset, and from Quarterly to other frequencies in the M4 → M3 dataset. Full result are shown in Table 11.

| Method | SE-LLM | AutoTimes | FPT | Dlinear | PatchTST | TimesNet | FEDformer | Informer | Reformer |
|---|---|---|---|---|---|---|---|---|---|
| M3→M4 | 13.024 | 13.036 | 13.125 | 15.337 | 13.228 | 14.553 | 15.047 | 19.047 | 14.092 |
| M4→M3 | 12.560 | 12.750 | 13.060 | 14.030 | 13.390 | 14.170 | 13.530 | 15.820 | 13.370 |

In the zero-shot experiments, we compared our approach (SE-LLM) with several state-of-the-art methods (see Table 3) and found that SE-LLM achieves competitive and slightly better performance in zero-shot generalization. Notably, when transferring from M3 to M4, the SMAPE was reduced by 0.1% compared to AutoTimes, and by 1.4% when transferring from M4 to M3. These results underscore the model's robustness in handling unseen data and diverse temporal patterns. Additionally, we examine domain generalization between the M3 and M4 datasets, where the temporal frequencies shift from monthly to weekly, daily, or hourly in the M3 → M4 transfer, and from quarterly to other frequencies in the M4 → M3 transfer. This domain adaptation capability, driven by the model's ability to learn a shared latent structure across datasets, plays a key role in improving performance in zero-shot scenarios. The results further verify the model's ability to generalize beyond the patterns seen during training.

## 4.2 ABLATION STUDY

**Ablation on Long-Term Forecasting.** Ablation experiments are conducted on the ECL and Traffic datasets. To reduce computational costs, we selected different lightweight LLMs for comparison, including GPT2 (Radford et al., 2019), BERT (Devlin et al., 2019), Opt-125m (Zhang et al., 2022), and Qwen2.5-0.5B (Yang et al., 2024a), which are commonly used in current research (Zhou et al., 2023; Liu et al., 2024c; 2025; Jin et al., 2024). The experimental results are shown in Table 4. Our baseline uses a cross-domain attention mechanism for multi-modal alignment (Chang et al., 2025; Jin et al., 2024; Liu et al., 2025). After replacing it with the TSCC framework and adding the Time-Adapter, the prediction accuracy of different LLMs has improved, demonstrating the effectiveness of the innovative methods in this paper. Comparative experiments show Qwen-0.5B yields lower prediction errors among LLMs. The ablation study on short-term forecasting are provided in the Appendix B.3 and Table 13.

Table 4: In the ablation experiments, we set the input length to 672 and the output length to 96, with forecast lengths from the set $\{96, 192, 336, 720\}$. Full results are provided in Table 12.

| LLM | GPT2 | | | | Bert | | | | Opt125m | | | | Qwen2.5-0.5B | | | |
|---|---|---|---|---|---|---|---|---|---|---|---|---|---|---|---|---|
| Dataset | ECL | | Traffic | | ECL | | Traffic | | ECL | | Traffic | | ECL | | Traffic | |
| Metrics | MSE | MAE | MSE | MAE | MSE | MAE | MSE | MAE | MSE | MAE | MSE | MAE | MSE | MAE | MSE | MAE |
| Baseline | 0.179 | 0.270 | 0.422 | 0.284 | 0.184 | 0.276 | 0.472 | 0.321 | 0.173 | 0.267 | 0.403 | 0.280 | 0.167 | 0.263 | 0.405 | 0.279 |
| +TSCC | 0.172 | 0.264 | 0.409 | 0.280 | 0.177 | 0.270 | 0.431 | 0.294 | 0.170 | 0.263 | 0.396 | 0.270 | 0.166 | 0.262 | 0.389 | 0.264 |
| +Time-Adapter | 0.166 | 0.258 | 0.406 | 0.279 | 0.167 | 0.260 | 0.412 | 0.282 | 0.163 | 0.257 | 0.391 | 0.266 | 0.161 | 0.255 | 0.386 | 0.261 |

**Ablation on TSCC Module.** The TSCC module features a sophisticated architecture. To better understand the contribution of each component, we perform a systematic ablation study, as summarized in Table 5. Specifically, we investigate the impact of removing or modifying key elements: (1) The AM-VAE module is removed to assess how the absence of simulated noisy data affects the modeling of temporal patterns. (2) The cross-domain attention mechanism is replaced with a simple linear layer for feature concatenation, thereby altering the approach to dynamic alignment and fusion across domains. (3) The gated attention mechanism is eliminated to examine the importance of capturing channel-wise dependencies. (4) Finally, the semantic space is substituted with a learnable parameter matrix, aiming to implicitly approximate the large language model's embedding without explicit semantic guidance. Experimental results show that the TSCC module provides reliable input to the large language model by effectively mining temporal patterns, injecting them into the semantic space, and employing an implicit prompting strategy. It is also worth noting that the effectiveness of the AM-VAE-based noise modeling is not strictly uniform across all datasets or model

configurations. For relatively stable temporal patterns, the benefit of explicitly modeling stochastic variations may be less pronounced, suggesting that this component is more effective in scenarios with stronger distributional shifts or irregular dynamics.

Table 5: Ablation study results on the ETTh1 dataset, where checkmarks indicate the modules employed, showing the results for different prediction horizons. All the strategies together yield the best results marked in red.

| Module Ablation | | | | 96 | | 192 | | 336 | | 720 | | Avg | |
|---|---|---|---|---|---|---|---|---|---|---|---|---|---|
| AM-VAE | Cross_Attn | Gated_Fusion | Semantic Space | MSE | MAE | MSE | MAE | MSE | MAE | MSE | MAE | MSE | MAE |
| | ✓ | ✓ | ✓ | 0.355 | 0.396 | 0.390 | 0.418 | 0.408 | 0.431 | 0.419 | 0.448 | 0.393 | 0.423 |
| ✓ | | ✓ | ✓ | 0.358 | 0.397 | 0.392 | 0.419 | 0.410 | 0.432 | 0.423 | 0.450 | 0.396 | 0.425 |
| ✓ | ✓ | | ✓ | 0.366 | 0.405 | 0.398 | 0.427 | 0.412 | 0.438 | 0.421 | 0.456 | 0.399 | 0.432 |
| ✓ | ✓ | ✓ | | 0.364 | 0.401 | 0.397 | 0.426 | 0.414 | 0.439 | 0.433 | 0.458 | 0.402 | 0.431 |
| ✓ | ✓ | ✓ | ✓ | 0.352 | 0.393 | 0.378 | 0.411 | 0.391 | 0.420 | 0.402 | 0.437 | 0.381 | 0.415 |

**Qualitative Analysis on TSCC Module.** In Fig. 5, **(a)** we extract a segment from the ETTh dataset and apply STL decomposition to obtain the trend, seasonal, and residual components. The local deviations and abrupt fluctuations in the residual component are treated as anomaly variations that our AM-VAE is designed to model. **(b)** STL decomposition reveals residual variations that are difficult to explain using traditional methods. The abrupt changes (marked by the red bar) are anomaly-related and challenging to characterize, while our AM-VAE module produces semantic components that respond to these segments. **(c)** The heatmap distribution of the variation-sensitive semantics is shown. **(d)** demonstrates noticeable overlap between some channels and the anomalous segments in the residuals, suggesting effective channel relationship mining for irregular patterns. In **(e-f)**, the t-SNE analysis shows that the residual anomaly-related distribution and variation-sensitive semantics form similar clusters. Seven tightly clustered points in the t-SNE projection indicate a potential alignment between the reconstructed semantics and the identified anomalies, suggesting that the reconstructed semantics are correlated with real anomaly-related variations, thus revealing latent patterns in the time series. In **(g)**, we show how variation-sensitive semantics are regularized along the time dimension and compared with anomaly patterns. **(h)** The linear distribution in the correlation map suggests a close relationship between the two, indicating that the model captures part of the statistical properties of the anomaly-related patterns.

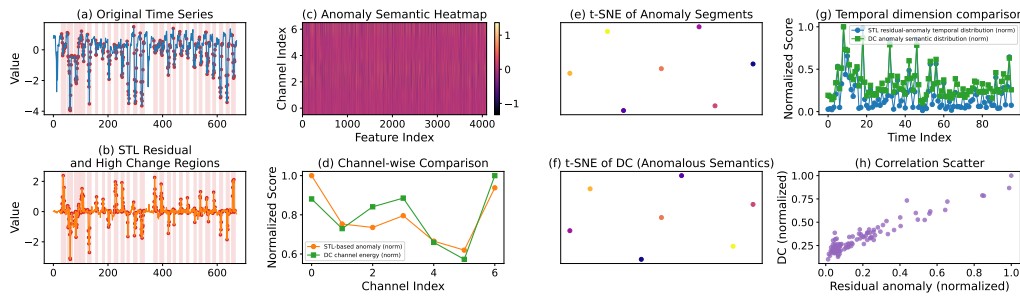

Figure 5: Qualitative analysis illustrating how the AM-VAE effectively models anomaly patterns.

**Ablation on Adapters.** Long-term forecasting should capture long-term dependencies, making the task more challenging. Therefore, to demonstrate the performance of Time-Adapter, we conducted a comparison with the LoRA (Hu et al., 2022; Liu et al., 2024c), and the results are presented in Table 6. The experimental baseline is the Qwen2.5 with the TSCC module. Experiments findings clearly demonstrate that LoRA lacks generalization capability to enhance forecasting performance. In contrast, Time-Adapter endows LLM with the ability to adapt to forecasting tasks.

**Ablation on LLM-based SOTA methods.** The modality gap between time-series and language data remains a challenging issue. To address this, we integrate TSCC and Time-Adapter as a plug-in into Time-LLM (Jin et al., 2024), AutoTimes (Liu et al., 2024c), and TimeCMA (Liu et al., 2025), evaluating their impact on the ETTh and ETTm datasets using the original code's default parameters. All other settings remain unchanged, with GPT-2 consistently serving as the LLM backbone. Results

Table 6: All reported results are averaged. Full results are provided in Table 14.

| Datasets | ETTh1 | | Weather | | Traffic | | ECL | | Solar | |
|---|---|---|---|---|---|---|---|---|---|---|
| Metrics | MSE | MAE | MSE | MAE | MSE | MAE | MSE | MAE | MSE | MAE |
| Baseline-TSCC | 0.381 | 0.415 | 0.242 | 0.287 | 0.389 | 0.264 | 0.166 | 0.262 | 0.201 | 0.251 |
| +LoRA | 0.396 | 0.425 | 0.240 | 0.278 | 0.390 | 0.268 | 0.163 | 0.257 | 0.202 | 0.253 |
| +Time-Adapter | 0.389 | 0.420 | 0.229 | 0.267 | 0.386 | 0.261 | 0.161 | 0.255 | 0.192 | 0.243 |

in Table 7 show consistent error reductions, with TimeCMA achieving a notable drop in MSE. Since TimeCMA uses LLMs only to generate outputs from textual time-series prompts without training the LLM, ablation studies on Time-Adapter are not feasible. These results demonstrate that our approach addresses a common limitation across existing frameworks.

Table 7: The forecasting horizons are in $\{96, 192, 336, 720\}$, and all experimental results presented are averages. Full results are provided in Table 15.

| | Time-LLM | | +TSCC | | +Time-Adapter | | AutoTimes | | +TSCC | | +Time-Adapter | | Time-CMA | | +TSCC | |
|---|---|---|---|---|---|---|---|---|---|---|---|---|---|---|---|---|
| | MSE | MAE | MSE | MAE | MSE | MAE | MSE | MAE | MSE | MAE | MSE | MAE | MSE | MAE | MSE | MAE |
| ETTh1 | 0.420 | 0.439 | 0.407 | 0.429 | 0.405 | 0.428 | 0.397 | 0.425 | 0.390 | 0.421 | 0.388 | 0.420 | 0.446 | 0.446 | 0.437 | 0.434 |
| ETTh2 | 0.370 | 0.404 | 0.368 | 0.401 | 0.364 | 0.399 | 0.367 | 0.410 | 0.363 | 0.406 | 0.357 | 0.407 | 0.416 | 0.429 | 0.379 | 0.403 |
| ETTm1 | 0.358 | 0.388 | 0.360 | 0.390 | 0.357 | 0.389 | 0.361 | 0.389 | 0.354 | 0.384 | 0.351 | 0.382 | 0.399 | 0.413 | 0.383 | 0.396 |
| ETTm2 | 0.262 | 0.324 | 0.267 | 0.328 | 0.266 | 0.326 | 0.274 | 0.329 | 0.269 | 0.325 | 0.268 | 0.323 | 0.294 | 0.337 | 0.285 | 0.347 |

## 4.3 EFFICIENCY ANALYSIS

We evaluate the computational efficiency of our approach. All experiments were conducted with a batch size of 256 and a sequence length of 672, under which the training and inference times for each method were recorded under the same hardware and software environment. The bar chart in Fig. 6 compares the training and inference speeds of SE-LLM with classical algorithms, all evaluated on the same dataset, highlighting the superior efficiency of our method. In addition, the bubble chart illustrates the relationship between training speed and MSE variations on the ECL dataset when our approach is applied with different LLMs, further demonstrating the trade-offs between computational speed and prediction accuracy.

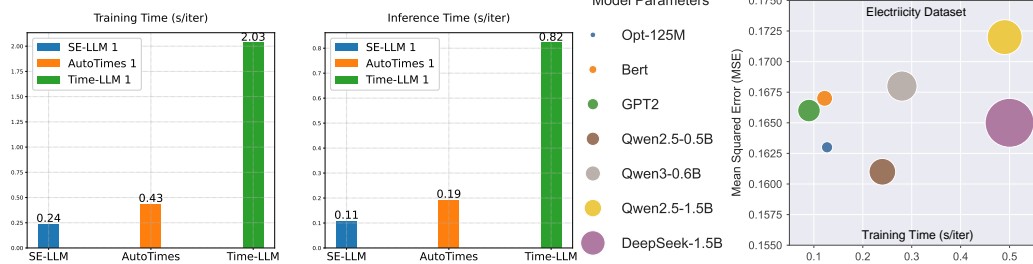

Figure 6: The size of the circles reflects the proportional relationship among the parameter counts of different LLMs.

## 5 CONCLUSION

This paper introduces SE-LLM, which addresses the inherent modality gap between LLMs and time series data. We explore the temporal and channel dependencies between temporal embeddings and the semantic space of LLMs. Token embeddings are enhanced with temporal-pattern semantic information, improving interpretability and unlocking LLMs' potential. Moreover, we incorporate a plugin module tailored for time-series forecasting into LLMs, which activates their ability to model temporal dependencies. Experimental results demonstrate that SE-LLM surpasses SOTA performance. Furthermore, embedding TSCC and Time-Adapter into existing frameworks also leads to performance improvements, shedding light on common challenges in leveraging LLMs for time-series forecasting. SE-LLM efficiently integrates temporal information and language models.

## 6 ACKNOWLEDGEMENTS

This research is supported by National Science and Technology Major Project (2022ZD0119202), National Science Fund for Distinguished Young Scholars (62125601), National Natural Science Foundation of China (62576031).

## 7 ETHICS STATEMENT

This work focuses on advancing the methodology of time series forecasting by enhancing large language models (LLMs) with semantic structures derived from temporal data patterns. Our approach does not involve human subjects, personal data, or sensitive information, and all experiments are conducted on publicly available time series datasets. The proposed SE-LLM framework is designed to improve model interpretability and computational efficiency without altering the core parameters of the pre-trained LLM. We do not foresee direct societal harms from this research.

## 8 REPRODUCIBILITY STATEMENT

Our code is available, and the detailed parameter configurations are provided in the released implementation. All datasets used in our experiments are publicly available. Most compared baselines are based on open-source implementations; for methods whose official implementations were not available at the time of our experiments, we followed the details reported in the original papers and used comparable experimental settings. These materials facilitate the reproducibility of our experimental results.

We did not conduct ablation experiments with very large language models, such as Llama-7B or larger, mainly because such models introduce substantially higher computational and memory costs than traditional time-series forecasting methods, which conflicts with the lightweight and practical deployment goals of this work. Although SE-LLM is built upon language-model backbones, we adopt smaller-scale LLMs to balance forecasting performance, efficiency, and practicality. Furthermore, we provide ablation studies with different LLM backbones in Table 13 and analyze computational efficiency in Section 4.3.

The discussion and limitation analyses are provided in Appendix C and Appendix D, respectively. The usage of large language models is discussed in Appendix E.

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

## A    APPENDIX FOR SE-LLM

### A.1    IMPLEMENTATION DETAILS OF THE TSCC PIPELINE

The pipeline consists of several interdependent components such as cross-attention, AM-VAE, Structural prior infusion, and gating fusion. Below, we provide detailed pseudocode for each component to improve readability and explain their roles within the pipeline.

**Cross-Modality Alignment.** The first step is to align semantic embeddings with temporal tokens using a cross-attention mechanism.

---
**Algorithm 2** Cross-Modality Alignment

---
1: **Input:** Temporal sequence **time_seq**, Semantic embeddings **word**.
2: **Output:** Temporally aligned semantic features **aligned_text**.
3: Compute query, key, and value matrices: {Project temporal tokens as queries; semantic tokens as keys/values.}
4: $\mathbf{Q} \leftarrow \text{Query}(\textbf{time\_seq})$
5: $\mathbf{K} \leftarrow \text{Key}(\textbf{word})$
6: $\mathbf{V} \leftarrow \text{Value}(\textbf{word})$
7: Calculate attention scores: {Measure similarity between temporal queries and semantic keys.}
8: $\textbf{attn\_scores} \leftarrow \frac{\mathbf{Q} \cdot \mathbf{K}^T}{\sqrt{d\_model}}$
9: Apply softmax to get attention weights: {Convert raw scores to normalized attention distribution.}
10: $\textbf{attn\_weights} \leftarrow \text{softmax}(\textbf{attn\_scores})$
11: Compute aligned semantic features: {Weighted aggregation of semantic values guided by temporal queries.}
12: $\textbf{aligned\_text} \leftarrow \textbf{attn\_weights} \cdot \mathbf{V}$

---

**Anomaly Pattern Modeling.** The AM-VAE module models latent anomaly patterns by estimating the underlying distribution and disentangling anomaly versus non-anomaly semantics.

---
**Algorithm 4** Structural Prior Infusion

---
1: **Input:** Temporal representation $\mathbf{H}$, semantic prototypes $\mathbf{S}$, anomaly feature $\mathbf{DC}$, and de-anomalized feature $\mathbf{DA}$.
2: **Output:** Structure-enhanced anomaly feature $\overline{\mathbf{DC}}$ and de-anomalized feature $\overline{\mathbf{DA}}$.
3: Compute the cross-correlation matrix: {Normalize temporal and semantic representations and compute their correlations.}
4: $\mathbf{M} \leftarrow \text{Norm}(\text{mean}(\mathbf{H})) \cdot \text{Norm}(\mathbf{S})^T$
5: Apply top-$k$ filtering: {Select the strongest temporal–semantic correlations as structural priors.}

6: $\overline{\mathbf{DA}} \leftarrow \mathbf{DA} \cdot \frac{1}{n} \sum_{i=1}^{n} \mathcal{P}(\mathbf{S}_i, m_{\text{top-}k}^i)$
7: $\overline{\mathbf{DC}} \leftarrow \mathbf{DC} \cdot \frac{1}{n} \sum_{i=1}^{n} \mathcal{P}(\mathbf{S}_i, m_{\text{top-}k}^i)$

---

---

**Algorithm 3** Anomaly Pattern Modeling via AM-VAE

---

1: **Input:** Cross-attention output $\mathbf{C}$ in the joint space.
2: **Output:** Anomaly semantic component $\mathbf{DC}$ and de-anomalized semantic component $\mathbf{DA}$.
3: Project the input into the latent space: {Encode the joint-space representation into latent features.}
4: $\mathbf{V} \leftarrow \text{Encoder}(\mathbf{C})$
5: Estimate the latent distribution: {Predict the mean and log-variance for variational inference.}
6: $\mu, \log \sigma^2 \leftarrow \text{Latent\_Distribution}(\mathbf{V})$
7: Apply the reparameterization trick: {Sample a differentiable latent code.}
8: $\mathbf{z} \leftarrow \mu + \epsilon \cdot \exp(0.5 \log \sigma^2)$
9: where $\epsilon \sim \mathcal{N}(\mathbf{0}, \mathbf{I})$.
10: Decode the latent variable: {Estimate the anomaly-related semantic component.}
11: $\mathbf{DC} \leftarrow \text{Decoder}(\mathbf{z})$
12: Compute the de-anomalized semantic component: {Remove the anomaly-related component from the original representation.}
13: $\mathbf{DA} \leftarrow \mathbf{C} - \mathbf{DC}$

---

**Gated Fusion.** A gate adaptively selects how much temporal vs. semantic information contributes to the fused representation.

---

**Algorithm 5** Gated Fusion of Temporal and Semantic Features

---

1: **Input:** Temporal feature $\textbf{time\_feat}$, Text feature $\textbf{text\_feat}$, Semantic embeddings $\textbf{word}$.
2: **Output:** Fused feature $\textbf{fused}$.
3: Compute similarity: {Match temporal features with the most relevant semantic tokens.}
4: $\textbf{sim\_matrix} \leftarrow \text{matmul}(\textbf{time\_feat}, \textbf{word}^T)$
5: Select top-k semantic tokens: {Keep only the most relevant semantic dimensions.}
6: $\textbf{topk\_idx} \leftarrow \text{topk}(\textbf{sim\_matrix}, k = 32)$
7: Enhance selected semantic features: {Aggregate top-k semantic cues.}
8: $\textbf{enhanced\_text} \leftarrow \text{mean}(\textbf{selected\_text}, \dim = 1)$
   {If a softmax operation is applied along a singleton dimension after aggregation, it degenerates to identity weighting and does not introduce additional adaptive weights.}
9: $\textbf{enhanced\_text} \leftarrow \text{softmax}(\textbf{enhanced\_text}, \dim = 1)$
10: Concatenate features: {Prepare inputs for gating network.}
11: $\textbf{combined} \leftarrow \text{concat}(\textbf{time\_feat}, \textbf{enhanced\_text})$
12: Compute gate values: {Predict adaptive weight between temporal and semantic paths.}
13: $\textbf{gate} \leftarrow \text{Gate\_Net}(\textbf{combined})$
14: Fuse features: {Final adaptive fusion.}
15: $\textbf{fused} \leftarrow \textbf{gate} \cdot \textbf{time\_feat} + (1 - \textbf{gate}) \cdot \textbf{text\_feat}$

---

**Final Model Fusion.** Combine anomaly and de-anomaly branches to produce the final enhanced representation. The final fused output is passed through a linear layer or subsequent predictor for forecasting.

---

**Algorithm 6** Final Fusion of Anomaly and De-Anomaly Features

---

1: **Input:** Fused anomaly output, Fused de-anomaly output.
2: **Output:** Final fused output $\textbf{fused\_output}$.
3: Combine both outputs: {Residual fusion capturing full semantic spectrum.}
4: $\textbf{fused\_output} \leftarrow \textbf{fused\_output\_da} + \textbf{fused\_output\_dc}$

---

## A.2 AUTOREGRESSIVE FORECASTING

In time series forecasting, autoregressive models offer the advantage of making predictions over arbitrary time horizons while being trained solely on sequences of fixed length, thereby significantly reducing computational overhead. These models generate future values by iteratively conditioning

on previously observed data points. This strictly sequential inference mechanism leverages the temporal continuity and intrinsic patterns present in historical observations. When these underlying patterns remain stable over time, autoregressive models typically produce reliable and consistent forecasts.

---

**Algorithm 7** Autoregressive Inference with Token Stride $\tau$

---

**Require:** Input $\mathbf{X} \in \mathbb{R}^{B \times L \times C}$, stride $\tau$, horizon $T$. Assume Model($\mathbf{X}$) outputs $\mathbf{O} \in \mathbb{R}^{B \times L \times C}$.
**Ensure:** Output $\hat{\mathbf{Y}} \in \mathbb{R}^{B \times T \times C}$.
1: $N \leftarrow \lfloor T/\tau \rfloor$
2: $r \leftarrow T - N \cdot \tau$
3: $K \leftarrow N + \mathbb{I}[r \neq 0] \; \{K = \lceil T/\tau \rceil\}$
4: $\mathcal{P} \leftarrow [\,] \; \{\text{store predicted chunks}\}$
5: **for** $j \leftarrow 1$ **to** $K$ **do**
6:    **if** $j > 1$ **then**
7:      $\mathbf{X} \leftarrow \text{Cat}(\mathbf{X}_{:,\,\tau+1:L,:},\; \mathbf{p}_{j-1})$
8:    **end if**
9:    $\mathbf{O} \leftarrow \text{Model}(\mathbf{X})$
10:    $\mathbf{p}_j \leftarrow \mathbf{O}_{:,\,L-\tau+1:L,:} \; \{\text{last } \tau \text{ steps}\}$
11:    append $\mathbf{p}_j$ to $\mathcal{P}$
12: **end for**
13: $\hat{\mathbf{Y}} \leftarrow \text{Cat}(\mathcal{P}) \; \{B \times (K \cdot \tau) \times C\}$
14: **if** $r \neq 0$ **then**
15:    $\hat{\mathbf{Y}} \leftarrow \hat{\mathbf{Y}}_{:,\,1:T,:} \; \{\text{truncate to } T\}$
16: **end if**
17: **return** $\hat{\mathbf{Y}}$

---

SE-LLM adopts an autoregressive inference strategy, inspired by the design of AutoTimes. Specifically, future values are generated by recursively feeding the model's prior outputs back into the input sequence as context for subsequent predictions. This iterative process enables long-horizon forecasting while maintaining coherence with the temporal dynamics of the series. The complete inference procedure is summarized in Algorithm 7.

## B   EXPERIMENT

### B.1   EXPERIMENTAL SETUP

**Evaluation Metrics.** Quantitative evaluation is essential for assessing time-series forecasting performance. In this study, we adopt standard metrics tailored to different forecasting scenarios. For long-term forecasting, we use Mean Squared Error (MSE) and Mean Absolute Error (MAE):

$$\text{MAE} = \frac{1}{N} \sum_{n=1}^{N} |Y_n - \hat{Y}_n|, \tag{9}$$

$$\text{MSE} = \frac{1}{N} \sum_{n=1}^{N} (Y_n - \hat{Y}_n)^2. \tag{10}$$

MAE reflects the average magnitude of forecast errors, while MSE penalizes larger deviations more heavily due to the squaring operation. For short-term and zero-shot forecasting on the M4 and M3 dataset, we additionally employ the following metrics:

Mean Absolute Percentage Error (MAPE):

$$\text{MAPE} = \frac{100}{N} \sum_{n=1}^{N} \frac{|Y_n - \hat{Y}_n|}{|Y_n|}, \tag{11}$$

Symmetric Mean Absolute Percentage Error (SMAPE):

$$\text{SMAPE} = \frac{200}{N} \sum_{n=1}^{N} \frac{|Y_n - \hat{Y}_n|}{|Y_n| + |\hat{Y}_n|}, \tag{12}$$

Mean Absolute Scaled Error (MASE):

$$\text{MASE} = \frac{1}{N} \sum_{n=1}^{N} \frac{|Y_n - \hat{Y}_n|}{\frac{1}{N-m} \sum_{j=m+1}^{N} |Y_j - Y_{j-m}|}. \tag{13}$$

Overall Weighted Average (OWA):

$$\text{OWA} = \frac{1}{2} \left( \frac{\text{SMAPE}}{\text{SMAPE}_{\text{Naïve2}}} + \frac{\text{MASE}}{\text{MASE}_{\text{Naïve2}}} \right). \tag{14}$$

MAPE and SMAPE provide scale-invariant percentage-based error assessments, with SMAPE mitigating issues related to small denominators. MASE offers a relative comparison against a naive seasonal benchmark. OWA combines MASE and SMAPE relative to the Naïve2 baseline, serving as a composite indicator of overall performance.

Table 8: The dataset includes detailed descriptions, with specifications such as dimensionality, the distribution of total time points across training, validation, and test partitions, the target range of future time points for prediction, and the temporal sampling rate.

| Tasks | Datasets | Dim | Forecast Horizon | Dataset Size | Frequency | Information |
|---|---|---|---|---|---|---|
| Long-Term Forecasting | ETTh1 | 7 | $\{96, 192, 336, 720\}$ | $\{8545, 2881, 2881\}$ | Hourly | Electricity |
| | Weather | 21 | $\{96, 192, 336, 720\}$ | $\{36792, 5271, 10540\}$ | 10min | Weather |
| | Traffic | 862 | $\{96, 192, 336, 720\}$ | $\{12185, 1757, 3509\}$ | Hourly | Transportation |
| | ECL | 321 | $\{96, 192, 336, 720\}$ | $\{18317, 2633, 5261\}$ | Hourly | Electricity |
| | Solar | 137 | $\{96, 192, 336, 720\}$ | $\{36601, 5161, 10417\}$ | 10min | Energy |
| Short-Term Forecasting and Zero-Shot Forecasting | M4-Yearly | 1 | 6 | $\{23000, 0, 23000\}$ | Yearly | Univariate |
| | M4-Quarterly | 1 | 8 | $\{24000, 0, 24000\}$ | Quarterly | Univariate |
| | M4-Monthly | 1 | 18 | $\{48000, 0, 48000\}$ | Monthly | Univariate |
| | M4-Weekly | 1 | 13 | $\{359, 0, 359\}$ | Weekly | Univariate |
| | M4-Daily | 1 | 14 | $\{4227, 0, 4227\}$ | Daily | Univariate |
| | M4-Hourly | 1 | 48 | $\{414, 0, 414\}$ | Hourly | Univariate |
| Zero-Shot Forecasting | M3-Yearly | 1 | 6 | $\{645, 0, 645\}$ | Yearly | Univariate |
| | M3-Quarterly | 1 | 8 | $\{756, 0, 756\}$ | Quarterly | Univariate |
| | M3-Monthly | 1 | 18 | $\{1428, 0, 1428\}$ | Monthly | Univariate |
| | M3-Others | 1 | 8 | $\{174, 0, 174\}$ | Others | Univariate |

**Datasets.** We adopted publicly available benchmarks in four research domains: long-term forecasting, short-term forecasting, zero-shot forecasting and in-context forecasting (Liu et al., 2024c). The experimental datasets comprised: ETTh (Zhou et al., 2021), ETTm (Zhou et al., 2021), Weather (Chen et al., 2021), Electricity (ECL) (Chen et al., 2021), Traffic (Chen et al., 2021), Solar (Lai et al., 2018), M3 (Wu et al., 2023), and M4 (Makridakis et al., 2020) datasets. The detailed introduction refers to the Table 8. Time series datasets consist of sequential data points collected or recorded at consistent time intervals (e.g., hourly, daily, monthly). Each entry includes a timestamp and one or more observed values. These datasets are used to train models that identify historical patterns to forecast future values. Common applications include predicting stock prices, energy demand, weather, sales, and website traffic.

**Baseline.** We compare SE-LLM with representative state-of-the-art forecasting methods, including: TimeMixer++ (Wang et al., 2025) and AutoTimes (Liu et al., 2024c) as the high-performance baselines for evaluation. The algorithms involved in all the experiments are TimeMOE (Shi et al., 2025), Time-CMA (Liu et al., 2025), TQNet (Lin et al., 2025), Time-LLM (Jin et al., 2024), LLM-TS (Chen et al., 2025), iTransformer (Liu et al., 2024b), DLinear (Zeng et al., 2023), Time-VLM (Zhong et al., 2025), S2IP-LLM (Pan et al., 2024a), PatchTST (Nie et al., 2023), TimesNet (Wu et al., 2023), FED-Former (Yang et al., 2024b), Informer (Zhou et al., 2021), Reformer (Kitaev et al., 2020), Koopa (Liu et al., 2023), and FPT (Zhou et al., 2023), some of which serve as supplementary references.

## B.2 COMPARATIVE RESULTS

**Long Term Forecasting.** The full results for long-term forecasting are shown in Table 9. On ETTh1, Traffic, ECL, and Solar, SE-LLM achieves the best or tied-best average performance. Overall, these

Table 9: The input length of the SE-LLM is 672. The forecast horizon is $\{96, 192, 336, 720\}$. The best results are in **red** and the second best are **blue**.

| Models | | SE-LLM | | TimeMixer++ 2025 | | TimeMOE 2025 | | TQNet 2025 | | Time-CMA 2025 | | LLM-TS 2025 | | AutoTimes-GPT2 2024 | | Time-LLM 2024 | | iTransformer 2024 | |
|---|---|---|---|---|---|---|---|---|---|---|---|---|---|---|---|---|---|---|---|
| Datasets | Metrics | MSE | MAE | MSE | MAE | MSE | MAE | MSE | MAE | MSE | MAE | MSE | MAE | MSE | MAE | MSE | MAE | MSE | MAE |
| ETTh1 | 96 | 0.352 | 0.393 | 0.375 | 0.404 | 0.349 | 0.379 | 0.370 | 0.403 | 0.366 | 0.396 | 0.403 | 0.420 | 0.360 | 0.397 | 0.380 | 0.412 | 0.387 | 0.418 |
|  | 192 | 0.378 | 0.411 | 0.415 | 0.436 | 0.395 | 0.413 | 0.417 | 0.430 | 0.401 | 0.420 | 0.440 | 0.441 | 0.391 | 0.419 | 0.405 | 0.422 | 0.416 | 0.437 |
|  | 336 | 0.391 | 0.420 | 0.442 | 0.451 | 0.447 | 0.453 | 0.452 | 0.457 | 0.412 | 0.431 | 0.471 | 0.457 | 0.408 | 0.432 | 0.422 | 0.433 | 0.434 | 0.450 |
|  | 720 | 0.402 | 0.437 | 0.482 | 0.482 | 0.457 | 0.462 | 0.505 | 0.510 | 0.406 | 0.427 | 0.503 | 0.487 | 0.429 | 0.452 | 0.430 | 0.459 | 0.447 | 0.473 |
|  | avg | 0.381 | 0.415 | 0.429 | 0.443 | 0.412 | 0.426 | 0.438 | 0.450 | 0.396 | 0.419 | 0.454 | 0.451 | 0.397 | 0.425 | 0.409 | 0.432 | 0.421 | 0.445 |
| Weather | 96 | 0.150 | 0.200 | 0.150 | 0.202 | 0.157 | 0.211 | 0.149 | 0.202 | 0.167 | 0.211 | 0.166 | 0.217 | 0.158 | 0.208 | 0.149 | 0.200 | 0.174 | 0.225 |
|  | 192 | 0.194 | 0.243 | 0.196 | 0.244 | 0.208 | 0.256 | 0.201 | 0.249 | 0.212 | 0.253 | 0.229 | 0.269 | 0.207 | 0.264 | 0.195 | 0.243 | 0.227 | 0.268 |
|  | 336 | 0.247 | 0.285 | 0.248 | 0.285 | 0.255 | 0.290 | 0.247 | 0.287 | 0.270 | 0.2927 | 0.278 | 0.302 | 0.262 | 0.298 | 0.245 | 0.282 | 0.290 | 0.309 |
|  | 720 | 0.323 | 0.339 | 0.316 | 0.332 | 0.405 | 0.397 | 0.312 | 0.332 | 0.350 | 0.348 | 0.354 | 0.351 | 0.342 | 0.353 | 0.318 | 0.338 | 0.374 | 0.360 |
|  | avg | 0.229 | 0.267 | 0.228 | 0.266 | 0.256 | 0.289 | 0.227 | 0.268 | 0.250 | 0.276 | 0.257 | 0.285 | 0.242 | 0.281 | 0.227 | 0.266 | 0.266 | 0.291 |
| Traffic | 96 | 0.350 | 0.243 | 0.367 | 0.266 | —— | —— | 0.366 | 0.255 | —— | —— | 0.587 | 0.315 | 0.369 | 0.257 | 0.373 | 0.280 | 0.353 | 0.259 |
|  | 192 | 0.373 | 0.254 | 0.379 | 0.364 | —— | —— | 0.392 | 0.276 | —— | —— | 0.612 | 0.326 | 0.394 | 0.268 | 0.390 | 0.288 | 0.373 | 0.267 |
|  | 336 | 0.390 | 0.263 | 0.417 | 0.301 | —— | —— | 0.400 | 0.280 | —— | —— | 0.634 | 0.338 | 0.413 | 0.278 | 0.407 | 0.299 | 0.386 | 0.275 |
|  | 720 | 0.429 | 0.285 | 0.453 | 0.314 | —— | —— | 0.442 | 0.300 | —— | —— | 0.640 | 0.351 | 0.449 | 0.299 | 0.438 | 0.310 | 0.425 | 0.296 |
|  | avg | 0.386 | 0.261 | 0.404 | 0.311 | —— | —— | 0.400 | 0.278 | —— | —— | 0.618 | 0.333 | 0.406 | 0.276 | 0.402 | 0.294 | 0.384 | 0.274 |
| ECL | 96 | 0.130 | 0.225 | 0.132 | 0.226 | —— | —— | 0.131 | 0.228 | 0.143 | 0.241 | 0.167 | 0.271 | 0.140 | 0.236 | 0.137 | 0.244 | 0.133 | 0.229 |
|  | 192 | 0.148 | 0.242 | 0.151 | 0.244 | —— | —— | 0.153 | 0.251 | 0.162 | 0.259 | 0.178 | 0.280 | 0.159 | 0.253 | 0.162 | 0.271 | 0.151 | 0.245 |
|  | 336 | 0.164 | 0.259 | 0.169 | 0.265 | —— | —— | 0.169 | 0.268 | 0.169 | 0.261 | 0.198 | 0.302 | 0.177 | 0.270 | 0.175 | 0.279 | 0.168 | 0.262 |
|  | 720 | 0.203 | 0.293 | 0.201 | 0.291 | —— | —— | 0.192 | 0.287 | 0.220 | 0.315 | 0.233 | 0.344 | 0.216 | 0.303 | 0.207 | 0.306 | 0.205 | 0.294 |
|  | avg | 0.161 | 0.255 | 0.163 | 0.257 | —— | —— | 0.161 | 0.259 | 0.174 | 0.269 | 0.173 | 0.266 | 0.173 | 0.268 | 0.170 | 0.275 | 0.164 | 0.258 |
| Solar | 96 | 0.165 | 0.220 | 0.177 | 0.235 | —— | —— | 0.197 | 0.244 | 0.202 | 0.222 | —— | —— | 0.179 | 0.220 | 0.224 | 0.289 | 0.183 | 0.265 |
|  | 192 | 0.183 | 0.235 | 0.190 | 0.249 | —— | —— | 0.230 | 0.274 | 0.216 | 0.240 | —— | —— | 0.198 | 0.236 | 0.244 | 0.289 | 0.205 | 0.283 |
|  | 336 | 0.199 | 0.248 | 0.199 | 0.262 | —— | —— | 0.244 | 0.281 | 0.236 | 0.255 | —— | —— | 0.213 | 0.252 | 0.225 | 0.291 | 0.224 | 0.299 |
|  | 720 | 0.219 | 0.266 | 0.209 | 0.276 | —— | —— | 0.247 | 0.280 | 0.252 | 0.264 | —— | —— | 0.239 | 0.277 | 0.243 | 0.301 | 0.239 | 0.316 |
|  | avg | 0.192 | 0.242 | 0.194 | 0.256 | —— | —— | 0.230 | 0.270 | 0.227 | 0.276 | —— | —— | 0.207 | 0.246 | 0.234 | 0.293 | 0.213 | 0.291 |

results demonstrate that SE-LLM achieves strong and consistent performance across diverse long-term forecasting benchmarks. Since the official implementation of TimeMixer++ was not publicly available at the time of our experiments, we reproduced it based on the details reported in the original paper and used the same input length and forecasting horizons for comparison. For methods whose results are adopted from prior work, we ensure that the reported settings are comparable in terms of forecasting horizons and evaluation metrics. For the remaining baselines, we re-ran the models under comparable settings and report the best results obtained across different input sequence lengths.

**Short Term Forecasting.** The full results for short-term series forecasting are shown in Table 10. The experimental results indicate that SE-LLM achieves the best performance in Yearly and Monthly patterns. From the lowest average error overall, SE-LLM achieve a lower SMAPE, signifying the best forecasting performance.

Table 10: The M4 dataset includes yearly, quarterly, monthly, weekly, daily, and hourly data, the short-term forecasts averaged across all M4 subsets.

| Models | | SE-LLM | FSCA 2025 | Time-VLM 2025 | AutoTimes 2024 | S2IP-LLM 2024 | Time-LLM 2024 | FPT 2024 | iTransformer 2024 |
|---|---|---|---|---|---|---|---|---|---|
| Yearly | SMAPE | 13.294 | 13.288 | 13.419 | 13.319 | 13.413 | 13.419 | 15.531 | 13.652 |
|  | MASE | 2.970 | 2.974 | 3.005 | 3.024 | 3.021 | 3.005 | 3.015 | 3.095 |
|  | OWA | 0.780 | 0.781 | 0.789 | 0.789 | 0.792 | 0.789 | 0.793 | 0.807 |
| Quarterly | SMAPE | 10.079 | 10.037 | 10.110 | 10.020 | 10.352 | 10.110 | 10.177 | 10.353 |
|  | MASE | 1.177 | 1.174 | 1.178 | 1.162 | 1.228 | 1.178 | 1.194 | 1.209 |
|  | OWA | 0.887 | 0.884 | 0.889 | 0.878 | 0.922 | 0.889 | 0.897 | 0.911 |
| Monthly | SMAPE | 12.618 | 12.762 | 12.980 | 12.696 | 12.995 | 12.980 | 12.894 | 13.079 |
|  | MASE | 0.931 | 0.947 | 0.963 | 0.936 | 0.970 | 0.963 | 0.956 | 0.974 |
|  | OWA | 0.875 | 0.897 | 0.903 | 0.880 | 0.910 | 0.903 | 0.897 | 0.911 |
| Others | SMAPE | 4.896 | 4.761 | 4.795 | 4.916 | 4.805 | 4.795 | 4.940 | 4.780 |
|  | MASE | 3.306 | 3.207 | 3.178 | 3.310 | 3.247 | 3.178 | 3.228 | 3.231 |
|  | OWA | 1.037 | 1.007 | 1.006 | 1.039 | 1.017 | 1.006 | 1.029 | 1.012 |
| Average | SMAPE | 11.778 | 11.828 | 11.894 | 11.831 | 12.021 | 11.983 | 11.991 | 12.142 |
|  | MASE | 1.578 | 1.580 | 1.592 | 1.585 | 1.612 | 1.595 | 1.600 | 1.631 |
|  | OWA | 0.847 | 0.850 | 0.855 | 0.850 | 0.857 | 0.859 | 0.861 | 0.874 |

**Zero-Shot Forecasting.** The full results for zero-shot forecasting are shown in Table 11. We adopt the zero-shot forecasting methodology from AutoTimes, where each experimental setup involves training a model on a designated source dataset and subsequently deploying it to generate forecasts on a target dataset. These zero-shot scenarios are executed across subsets categorized by their sampling frequencies, each exhibiting distinct distributional properties. Overall, SE-LLM achieves the best performance.

Table 11: The experimental results are based on the generalization of consistent temporal patterns. For non-matching patterns, we evaluate frequency transfers from Monthly to Weekly/Daily/Hourly in the M3 → M4 dataset, and from Quarterly to other frequencies in the M4 → M3 dataset.

| Method | | SE-LLM | AutoTimes | FPT | Dlinear | PatchTST | TimesNet | FEDformer | Informer | Reformer |
|---|---|---|---|---|---|---|---|---|---|---|
| | Yearly | 13.706 | 13.708 | 13.740 | 14.193 | 13.966 | 15.655 | 13.887 | 18.542 | 15.662 |
| | Quarterly | 10.718 | 10.742 | 10.787 | 18.856 | 10.929 | 11.877 | 11.513 | 16.907 | 11.051 |
| M3→M4 | Monthly | 14.553 | 14.558 | 14.630 | 14.765 | 14.664 | 16.165 | 18.154 | 23.454 | 15.604 |
| | Others | 6.269 | 6.259 | 7.081 | 9.194 | 7.087 | 6.863 | 7.529 | 7.348 | 7.001 |
| | Avg | 13.024 | 13.036 | 13.125 | 15.337 | 13.228 | 14.553 | 15.047 | 19.047 | 14.092 |
| | Yearly | 15.692 | 15.731 | 16.420 | 17.430 | 15.990 | 18.750 | 16.000 | 19.700 | 16.030 |
| | Quarterly | 9.011 | 9.350 | 10.130 | 9.740 | 9.620 | 12.260 | 9.480 | 13.000 | 9.760 |
| M4→M3 | Monthly | 13.978 | 14.060 | 14.100 | 15.650 | 14.710 | 14.010 | 15.120 | 15.910 | 14.800 |
| | Others | 4.872 | 5.790 | 4.810 | 6.810 | 9.440 | 6.880 | 8.940 | 13.030 | 7.530 |
| | Avg | 12.560 | 12.750 | 13.060 | 14.030 | 13.390 | 14.170 | 13.530 | 15.820 | 13.370 |

## B.3 ABLATION STUDY

**Ablation on Long-Term Forecasting.** The full results of the innovation methods are shown in Table 12. SE-LLM uses cross-domain attention alignment (Jin et al., 2024; Chang et al., 2025; Liu et al., 2025) as the baseline. After replacing it with TSCC, errors were reduced across different LLMs. To make LLMs more suitable for processing time-series data, we incorporated a Time-Adapter, which further improved the model's predictive performance. The ablation results demonstrate the effectiveness of our innovations.

Table 12: In the ablation experiments, we set the input length to 672 and the output length to 96, with forecast lengths from the set $\{96, 192, 336, 720\}$. The values in **bold** represent the average results, while red indicates the best performance, followed by blue.

| LLM | | GPT2 | | | | Bert | | | | Opt125m | | | | Qwen2.5-0.5B | | | |
|---|---|---|---|---|---|---|---|---|---|---|---|---|---|---|---|---|---|
| Dataset | | ECL | | Traffic | | ECL | | Traffic | | ECL | | Traffic | | ECL | | Traffic | |
| Metrics | | MSE | MAE | MSE | MAE | MSE | MAE | MSE | MAE | MSE | MAE | MSE | MAE | MSE | MAE | MSE | MAE |
| Baseline | 96 | 0.145 | 0.240 | 0.387 | 0.267 | 0.150 | 0.245 | 0.440 | 0.304 | 0.140 | 0.237 | 0.368 | 0.258 | 0.135 | 0.233 | 0.373 | 0.263 |
| | 192 | 0.165 | 0.257 | 0.410 | 0.277 | 0.168 | 0.262 | 0.459 | 0.314 | 0.158 | 0.254 | 0.390 | 0.268 | 0.152 | 0.249 | 0.394 | 0.272 |
| | 336 | 0.183 | 0.275 | 0.427 | 0.286 | 0.187 | 0.281 | 0.476 | 0.322 | 0.176 | 0.272 | 0.407 | 0.292 | 0.170 | 0.267 | 0.409 | 0.280 |
| | 720 | 0.224 | 0.309 | 0.463 | 0.306 | 0.230 | 0.316 | 0.514 | 0.344 | 0.216 | 0.305 | 0.445 | 0.300 | 0.212 | 0.303 | 0.444 | 0.300 |
| | Avg | **0.179** | **0.270** | **0.422** | **0.284** | **0.184** | **0.276** | **0.472** | **0.321** | **0.173** | **0.267** | **0.403** | **0.280** | **0.167** | **0.263** | **0.405** | **0.279** |
| +TSCC | 96 | 0.138 | 0.233 | 0.368 | 0.257 | 0.144 | 0.240 | 0.395 | 0.277 | 0.136 | 0.232 | 0.361 | 0.251 | 0.134 | 0.231 | 0.356 | 0.247 |
| | 192 | 0.157 | 0.250 | 0.393 | 0.269 | 0.161 | 0.256 | 0.417 | 0.286 | 0.154 | 0.249 | 0.383 | 0.263 | 0.152 | 0.248 | 0.377 | 0.257 |
| | 336 | 0.176 | 0.269 | 0.414 | 0.281 | 0.181 | 0.274 | 0.436 | 0.295 | 0.173 | 0.267 | 0.401 | 0.271 | 0.169 | 0.266 | 0.393 | 0.266 |
| | 720 | 0.218 | 0.305 | 0.461 | 0.312 | 0.223 | 0.309 | 0.474 | 0.316 | 0.215 | 0.303 | 0.439 | 0.294 | 0.210 | 0.301 | 0.431 | 0.287 |
| | Avg | **0.172** | **0.264** | **0.409** | **0.280** | **0.177** | **0.270** | **0.431** | **0.294** | **0.170** | **0.263** | **0.396** | **0.270** | **0.166** | **0.262** | **0.389** | **0.264** |
| +Time-Adapter | 96 | 0.133 | 0.227 | 0.363 | 0.255 | 0.135 | 0.230 | 0.378 | 0.266 | 0.132 | 0.227 | 0.354 | 0.247 | 0.130 | 0.225 | 0.350 | 0.243 |
| | 192 | 0.152 | 0.244 | 0.388 | 0.267 | 0.152 | 0.247 | 0.400 | 0.275 | 0.149 | 0.243 | 0.378 | 0.258 | 0.148 | 0.242 | 0.373 | 0.254 |
| | 336 | 0.169 | 0.262 | 0.411 | 0.281 | 0.170 | 0.265 | 0.417 | 0.283 | 0.166 | 0.261 | 0.396 | 0.267 | 0.164 | 0.259 | 0.390 | 0.263 |
| | 720 | 0.211 | 0.299 | 0.461 | 0.314 | 0.209 | 0.298 | 0.454 | 0.303 | 0.204 | 0.295 | 0.434 | 0.291 | 0.203 | 0.293 | 0.429 | 0.285 |
| | Avg | **0.166** | **0.258** | **0.406** | **0.279** | **0.167** | **0.260** | **0.412** | **0.282** | **0.163** | **0.257** | **0.391** | **0.266** | **0.161** | **0.255** | **0.386** | **0.261** |

**Ablation on Short-Term Forecasting.** The ablation experiments for short-term forecasting are presented in the Table 13. We compared the performance by replacing different Large Language Models (LLMs) with SE-LLM on the M4 dataset. The findings indicate that LLama exhibits lower errors, followed by Qwen2.5-0.5B.

**Ablation on Adapters.** We use the best-performing Qwen2.5-0.5B model as the baseline and evaluate the proposed Time-Adapter's performance on time-series forecasting tasks. The results are compared to the LoRA adapter. The LoRA parameter configuration is kept consistent with the Time-Adapter, with a rank of 8 and the full ablation results are presented in Table 14. The comparison protocol involves evaluating both adapters across several time-series datasets, specifically focusing on the performance metrics for each dataset. Our findings reveal that the LoRA adapter shows improvements in only the weather and ECL datasets, whereas the Time-Adapter consistently enhances performance across all datasets. In particular, the error reduction is most notable in the weather dataset, where the Time-Adapter achieves a 5.4% decrease in MSE compared to the baseline. This suggests that the Time-Adapter provides a more generalized performance boost across diverse time-series data, as compared to LoRA.

Table 13: The first column represents different LLMs replaced based on SE-LLM. "Others" is obtained by averaging the results from the weekly, daily, and hourly periods.

| | LLM | GPT2 | Bert | Opt-125m | Qwen2.5 | Llama |
|---|---|---|---|---|---|---|
| Year | sMAPE | 13.843 | 15.659 | 14.028 | 13.396 | 13.294 |
| | MASE | 3.199 | 3.674 | 3.198 | 3.002 | 2.970 |
| | OWA | 0.826 | 0.941 | 0.832 | 0.788 | 0.780 |
| Quarterly | sMAPE | 10.300 | 12.237 | 10.320 | 10.168 | 10.079 |
| | MASE | 1.198 | 1.573 | 1.207 | 1.182 | 1.177 |
| | OWA | 0.904 | 1.129 | 0.909 | 0.893 | 0.887 |
| Monthly | sMAPE | 12.935 | 14.376 | 13.273 | 12.738 | 12.618 |
| | MASE | 0.957 | 1.118 | 1.021 | 0.938 | 0.931 |
| | OWA | 0.898 | 1.024 | 0.940 | 0.883 | 0.875 |
| Others | sMAPE | 5.061 | 6.358 | 5.191 | 4.998 | 4.896 |
| | MASE | 3.468 | 4.367 | 3.507 | 3.417 | 3.306 |
| | OWA | 1.079 | 1.358 | 1.099 | 1.065 | 1.037 |
| Average | sMAPE | **12.121** | **13.757** | **12.334** | **11.886** | **11.778** |
| | MASE | **1.660** | **1.977** | **1.691** | **1.595** | **1.578** |
| | OWA | **0.881** | **1.024** | **0.897** | **0.855** | **0.847** |

Table 14: Time-Adapter compared with LoRA. The input length is 672 and forecasting horizons were set to $\{96, 192, 336, 720\}$.

| Datasets | | ETTh1 | | Weather | | Traffic | | ECL | | Solar | |
|---|---|---|---|---|---|---|---|---|---|---|---|
| Metrics | | MSE | MAE | MSE | MAE | MSE | MAE | MSE | MAE | MSE | MAE |
| TSCC | 96 | 0.352 | 0.393 | 0.162 | 0.245 | 0.356 | 0.247 | 0.134 | 0.231 | 0.175 | 0.230 |
| | 192 | 0.378 | 0.411 | 0.209 | 0.259 | 0.377 | 0.257 | 0.152 | 0.248 | 0.194 | 0.245 |
| | 336 | 0.391 | 0.420 | 0.263 | 0.299 | 0.393 | 0.266 | 0.169 | 0.266 | 0.209 | 0.257 |
| | 720 | 0.402 | 0.437 | 0.332 | 0.346 | 0.431 | 0.287 | 0.210 | 0.301 | 0.227 | 0.270 |
| | Avg | **0.381** | **0.415** | **0.242** | **0.287** | **0.389** | **0.264** | **0.166** | **0.262** | **0.201** | **0.251** |
| +LoRA | 96 | 0.362 | 0.399 | 0.157 | 0.210 | 0.357 | 0.251 | 0.132 | 0.228 | 0.173 | 0.228 |
| | 192 | 0.393 | 0.420 | 0.204 | 0.254 | 0.378 | 0.260 | 0.149 | 0.244 | 0.194 | 0.245 |
| | 336 | 0.410 | 0.432 | 0.260 | 0.297 | 0.394 | 0.269 | 0.166 | 0.261 | 0.210 | 0.260 |
| | 720 | 0.418 | 0.447 | 0.339 | 0.351 | 0.430 | 0.290 | 0.205 | 0.295 | 0.229 | 0.277 |
| | Avg | **0.396** | **0.425** | **0.240** | **0.278** | **0.390** | **0.268** | **0.163** | **0.257** | **0.202** | **0.253** |
| +Time-Adapter | 96 | 0.355 | 0.398 | 0.150 | 0.200 | 0.350 | 0.243 | 0.130 | 0.225 | 0.165 | 0.220 |
| | 192 | 0.389 | 0.421 | 0.194 | 0.243 | 0.373 | 0.254 | 0.148 | 0.242 | 0.183 | 0.236 |
| | 336 | 0.409 | 0.435 | 0.247 | 0.285 | 0.390 | 0.263 | 0.164 | 0.259 | 0.199 | 0.249 |
| | 720 | 0.418 | 0.448 | 0.323 | 0.339 | 0.429 | 0.285 | 0.203 | 0.293 | 0.219 | 0.266 |
| | Avg | **0.393** | **0.426** | **0.229** | **0.267** | **0.386** | **0.261** | **0.161** | **0.255** | **0.192** | **0.243** |

**Ablation on LLM-based SOTA methods.** In this paper, we propose TSCC and Time-Adapter to enhance the interpretability of LLMs. Furthermore, we integrate our methods into Time-LLM, AutoTimes, and Time-CMA. However, due to inherent algorithmic constraints, Time-CMA relies solely on the LLM to extract textual descriptions of time-series data and thus cannot incorporate Time-Adapter. In Table 15, we present the performance enhancement achieved by our method in long-term forecasting compared to current research. It is worth noting that the original autoregressive forecasting paradigm of AutoTimes is not well suited for the ETT dataset. Therefore, we make several modifications in our implementation. Specifically, the LLM component is instantiated with GPT-2, a multi-step forecasting strategy is adopted instead of autoregressive prediction, and temporal positional encodings are integrated at the input level.

# C  DISCUSSION

The proposed TSCC and Time-Adapter modules are designed to explicitly bridge the gap between large language models and time series forecasting from a modeling perspective. Rather than directly transferring language models to temporal data through prompt engineering or token-level alignment alone, TSCC focuses on enriching semantic representations with structured temporal correlations, while the Time-Adapter addresses the mismatch in temporal dependency modeling by explicitly capturing both long-term and short-term dynamics. Experimental results demonstrate that the general knowledge encoded in pre-trained language models is insufficient for structured temporal reasoning unless it is coupled with mechanisms that explicitly encode temporal structure, dependencies, and

Table 15: The results were conducted in the ETTh1, ETTh2, ETTm1, and ETTm2 datasets. The **Baseline** represents our reproduced results, which differ from those in the original paper. Under the same parameters, we added TSCC and Time-Adapter. The forecasting horizons were set to {96, 192, 336, 720}.

| Methods | | Time-LLM | | TSCC | | Time-Adapter | | AutoTimes | | TSCC | | Time-Adapter | | Time-CMA | | TSCC | |
|---|---|---|---|---|---|---|---|---|---|---|---|---|---|---|---|---|---|
| Metrics | | MSE | MAE | MSE | MAE | MSE | MAE | MSE | MAE | MSE | MAE | MSE | MAE | MSE | MAE | MSE | MAE |
| ETTh1 | 96 | 0.384 | 0.411 | 0.373 | 0.401 | 0.368 | 0.400 | 0.360 | 0.397 | 0.354 | 0.394 | 0.352 | 0.393 | 0.393 | 0.414 | 0.377 | 0.396 |
| | 192 | 0.414 | 0.430 | 0.403 | 0.425 | 0.401 | 0.421 | 0.391 | 0.419 | 0.385 | 0.415 | 0.383 | 0.414 | 0.435 | 0.437 | 0.429 | 0.424 |
| | 336 | 0.433 | 0.448 | 0.414 | 0.430 | 0.412 | 0.430 | 0.408 | 0.432 | 0.401 | 0.427 | 0.399 | 0.425 | 0.472 | 0.453 | 0.468 | 0.444 |
| | 720 | 0.448 | 0.468 | 0.436 | 0.460 | 0.439 | 0.459 | 0.429 | 0.452 | 0.420 | 0.446 | 0.417 | 0.446 | 0.482 | 0.479 | 0.474 | 0.471 |
| | Avg | **0.420** | **0.439** | **0.407** | **0.429** | **0.405** | **0.428** | **0.397** | **0.425** | **0.390** | **0.421** | **0.388** | **0.420** | **0.446** | **0.446** | **0.437** | **0.434** |
| ETTh2 | 96 | 0.293 | 0.349 | 0.287 | 0.352 | 0.294 | 0.347 | 0.288 | 0.352 | 0.288 | 0.352 | 0.285 | 0.377 | 0.339 | 0.378 | 0.299 | 0.347 |
| | 192 | 0.379 | 0.400 | 0.367 | 0.393 | 0.370 | 0.400 | 0.351 | 0.395 | 0.350 | 0.393 | 0.351 | 0.390 | 0.421 | 0.425 | 0.377 | 0.395 |
| | 336 | 0.381 | 0.414 | 0.390 | 0.414 | 0.373 | 0.402 | 0.383 | 0.424 | 0.378 | 0.419 | 0.376 | 0.413 | 0.452 | 0.452 | 0.415 | 0.426 |
| | 720 | 0.428 | 0.452 | 0.426 | 0.446 | 0.418 | 0.445 | 0.447 | 0.468 | 0.435 | 0.461 | 0.416 | 0.446 | 0.452 | 0.462 | 0.424 | 0.443 |
| | Avg | **0.370** | **0.404** | **0.368** | **0.401** | **0.364** | **0.399** | **0.367** | **0.410** | **0.363** | **0.406** | **0.357** | **0.407** | **0.416** | **0.429** | **0.379** | **0.403** |
| ETTm1 | 96 | 0.300 | 0.356 | 0.302 | 0.354 | 0.303 | 0.354 | 0.291 | 0.347 | 0.286 | 0.341 | 0.285 | 0.340 | 0.340 | 0.380 | 0.311 | 0.352 |
| | 192 | 0.340 | 0.378 | 0.341 | 0.378 | 0.345 | 0.376 | 0.338 | 0.376 | 0.334 | 0.371 | 0.332 | 0.369 | 0.379 | 0.407 | 0.360 | 0.381 |
| | 336 | 0.370 | 0.395 | 0.373 | 0.399 | 0.366 | 0.394 | 0.376 | 0.398 | 0.370 | 0.394 | 0.366 | 0.392 | 0.405 | 0.413 | 0.396 | 0.405 |
| | 720 | 0.422 | 0.428 | 0.422 | 0.427 | 0.414 | 0.428 | 0.438 | 0.433 | 0.427 | 0.429 | 0.421 | 0.426 | 0.472 | 0.450 | 0.464 | 0.444 |
| | Avg | **0.358** | **0.389** | **0.360** | **0.390** | **0.357** | **0.388** | **0.361** | **0.389** | **0.354** | **0.384** | **0.351** | **0.382** | **0.399** | **0.413** | **0.383** | **0.396** |
| ETTm2 | 96 | 0.172 | 0.265 | 0.180 | 0.270 | 0.178 | 0.268 | 0.176 | 0.264 | 0.174 | 0.261 | 0.173 | 0.262 | 0.191 | 0.274 | 0.173 | 0.256 |
| | 192 | 0.232 | 0.306 | 0.247 | 0.314 | 0.229 | 0.303 | 0.238 | 0.307 | 0.236 | 0.303 | 0.234 | 0.302 | 0.257 | 0.316 | 0.242 | 0.301 |
| | 336 | 0.281 | 0.339 | 0.276 | 0.335 | 0.291 | 0.342 | 0.298 | 0.346 | 0.291 | 0.341 | 0.291 | 0.337 | 0.313 | 0.350 | 0.307 | 0.323 |
| | 720 | 0.362 | 0.387 | 0.366 | 0.391 | 0.364 | 0.391 | 0.382 | 0.400 | 0.373 | 0.394 | 0.373 | 0.389 | 0.416 | 0.407 | 0.416 | 0.407 |
| | Avg | **0.262** | **0.324** | **0.267** | **0.328** | **0.266** | **0.326** | **0.274** | **0.329** | **0.269** | **0.325** | **0.268** | **0.323** | **0.294** | **0.337** | **0.285** | **0.322** |

variability. From this perspective, our work suggests a principled direction for future research on aligning foundation models with structured non-linguistic data, where adaptation should be guided by the intrinsic properties of the target domain rather than relying on superficial modality conversion.

# D LIMITATION

We provide further clarification on the limitations of SE-LLM. Firstly, in short-term and zero-shot forecasting using the LLama model, Time-Adapter was not employed in all instances due to computational constraints, which contradicts the low computational cost objective of this paper. As a result, Time-Adapter was excluded from these experiments. In fact, incorporating Time-Adapter in specific layers would likely lead to performance improvements. However, due to concerns about computational costs, we did not pursue this avenue in the current study. Additionally, LLama was not considered for long-term forecasting due to its generally poor performance and high computational demands.

# E THE USE OF LLMS

This paper employs a large language model (LLM) to check for spelling errors, correct grammar, and refine the language.

