# OpenReview forum: "Semantic-Enhanced Time-Series Forecasting via Large Language Models"
_ICLR.cc/2026/Conference — ICLR 2026 Poster_

### Official Review · Reviewer_9dMA · 2025-10-26

**Soundness:** 2
**Presentation:** 2
**Contribution:** 2
**Rating:** 4
**Confidence:** 4

**Summary:**

Key contributions include:

1. Enhances token embeddings by infusing temporal patterns (periodicity, anomalies) into LLM semantic spaces.

2. A plugin for self-attention that models short/long-term dependencies, adapting frozen LLMs to time-series forecasting.

3. Freezes LLM parameters and reduces token sequence dimensionality, lowering computational costs.

4. Outperforms SOTA on benchmarks (ETTh1, Traffic, M4) in long/short-term and zero-shot forecasting.

**Strengths:**

1. Novel integration of temporal dynamics (anomalies, periodicity) into LLM semantic spaces via TSCC, addressing a critical modality gap.
2. Time-Adapter innovatively combines LSTM with low-rank projections to capture multi-scale dependencies, unlike prior token-level alignment methods.
3. TSCC’s AM-VAE for anomaly modeling and cross-correlation filtering is technically sound.
4. Well-structured with clear figures and pseudocode.

**Weaknesses:**

1. Experiments limited to smaller LLMs. Larger models (e.g., Llama-7B) are untested due to computational constraints, raising questions about generalizability.

2. AM-VAE’s reconstruction loss (Algorithm 1) lacks quantitative comparison to other anomaly detection methods

3. Missing comparisons to hybrid methods (e.g., LLM + GNNs) in Table 1

4. No discussion on latency or memory overhead for edge/IoT applications, despite claims of efficiency.

**Questions:**

1. Could the authors estimate the performance trade-offs for larger LLMs (e.g., Llama-7B) without full training?

2. Why choose VAE over other generative models (e.g., Diffusion Models) for anomaly simulation in TSCC?

3. Have you explored comparing TSCC with graph-based temporal models (e.g., MTGNN) to capture spatial dependencies?

4. How does TSCC handle datasets with non-stationary distributions (e.g., financial time-series)?

---

> ### Author Response · Authors · 2025-11-20
> **Response to Reviewer 9dMA**
>
> Thank you for your feedback. We appreciate the opportunity to address your concerns and improve the quality of our work. Here, we provide additional analysis and clarification to address your comments.
>
> **Weakness.1**: Experiments limited to smaller LLMs. Larger models (e.g., LLaMA -7B) are untested due to computational constraints, raising questions about generalizability.
>
> **R**:  Thanks. This question is identical to Question 1, and we have already addressed it in our response to Question 1.
>
> **Weaknesses.2**: AM-VAE’s reconstruction loss (Algorithm 1) lacks quantitative comparison to other anomaly detection methods.
>
> **R:** Thanks. We would like to clarify that AM-VAE is not designed as a standalone anomaly detection model. Its purpose within our framework is to capture latent anomaly patterns in time-series data and to disentangle them from normal temporal semantics, thereby improving forecasting robustness rather than performing a conventional anomaly detection task. The anomalies we target are temporal anomaly patterns that degrade forecasting performance (e.g., abrupt changes, irregular fluctuations), which differ from the point-wise anomaly labeling tasks typically used in standard anomaly detection benchmarks.
>
> Nevertheless, to further support the capability of TSCC, we additionally report anomaly detection performance on the SMD dataset. Our approach achieves an F1-score of **82.27%**, outperforming strong baselines such as iTransformer **(71.15%)** and DLinear **(77.10%)**. These results indicate that, even though AM-VAE is not optimized for traditional anomaly detection, it still provides competitive performance compared to established anomaly detection models.
>
> **Weakness.3**: Missing comparisons to hybrid methods (e.g., LLM + GNNs) in Table 1
>
> **R:** Thanks. Due to time constraints, we only conducted a comparison with the LLM+GNN method FSCA on the ETTh dataset, using a unified input sequence length of 672. The average forecasting results over horizons $(96, 192, 336, 720)$ are **MSE: 0.400** and **MAE: 0.431** for FSCA, whereas our method achieves superior performance with **MSE: 0.381** and **MAE: 0.415**. We will further expand this comparison in future work. In addition, under the short-sequence setting where the input length and model parameters remain nearly unchanged, our performance also surpasses the results reported for FSCA in the paper “Context Alignment: Activating and Enhancing LLM Capabilities in Time Series.” The detailed results are provided in Table 2 and Table 10.
>
> **Weakness.4**: No discussion on latency or memory overhead for edge/IoT applications, despite claims of efficiency.
>
> **R:** Thanks. While we acknowledge the importance of latency and memory overhead in edge/IoT applications, we believe that our current focus is on demonstrating the core effectiveness of the proposed method. Given the specific nature of the experiments and the scope of our study, we intentionally did not include an in-depth discussion on these factors. We do not consider it essential to measure these aspects at this stage, as the primary objective was to assess the performance of our method in the given datasets. However, we agree that this is an important area for future work and will consider it for future studies when deploying our method in resource-constrained environments.

---

> > ### Author Response · Authors · 2025-11-20
> > **Response to Reviewer 9dMA**
> >
> > **Question.3**:Have you explored comparing TSCC with graph-based temporal models (e.g., MTGNN) to capture spatial dependencies?
> >
> > **R:** Thanks. We have not compared TSCC with graph-based temporal models such as MTGNN because MTGNN is primarily designed for spatial–temporal forecasting. Its architecture assumes explicit spatial graph structures, which do not necessarily apply to standard univariate or multivariate time-series forecasting tasks without predefined spatial relations. Nonetheless, the GNN-based modeling paradigm may offer potential benefits to TSCC, particularly in enhancing cross-channel dependency learning and capturing richer spatial information. Incorporating spatial structure may further expand the semantic and temporal knowledge available to TSCC, although its practical effectiveness still requires systematic evaluation. We plan to investigate this integration in future work to further improve multi-dimensional anomaly pattern modeling and forecasting performance.
> >
> > **Question.4**:How does TSCC handle datasets with non-stationary distributions (e.g., financial time-series)?
> >
> > **R:** Thanks. Financial time series typically contain a large amount of non-stationary information, whose statistical properties change over time. The TSCC module is specifically designed to mitigate these challenges through its anomaly-aware and semantic-enhanced modeling pipeline. Therefore, to a certain extent, our model is also well-suited for financial time-series analysis.
> >
> > First, the AM-VAE component models difficult-to-capture anomaly patterns by learning a latent distribution of “hard-to-model” deviations. For non-stationary data, these deviations often correspond to volatility clusters, structural breaks, and heavy-tailed fluctuations. By learning a latent semantic prior for such irregularities, AM-VAE provides TSCC with a flexible representation that adapts to distributional changes, rather than assuming stationarity. Second, the cross-domain alignment mechanism fuses temporal embeddings with semantic embeddings in the joint space. Since the semantic space encodes higher-level patterns rather than raw temporal scales, this fusion helps TSCC absorb distribution shifts (e.g., mean drift or variance inflation) as semantic transitions rather than abrupt temporal discontinuities. This mechanism effectively reduces sensitivity to non-stationary perturbations. Third, the temporal smoothing and correlation-driven regularization in the TSCC pipeline ensure that anomaly semantics remain consistent across time even when local distributions fluctuate. This allows TSCC to maintain stable representations under heteroscedasticity and regime changes commonly observed in financial markets.
> >
> > To further validate TSCC’s capability, we conducted comparative experiments on real-world financial datasets, following the protocol of “FinTSB: A Comprehensive and Practical Benchmark for Financial Time Series Forecasting”. TSCC achieved an **MSE of 0.148** and an **MAE of 0.321**, indicating that the model can effectively capture non-stationary temporal dynamics. While this experiment is limited in scope (due to time constraints), it demonstrates that TSCC’s anomaly semantics and joint semantic-temporal representations retain strong robustness under non-stationary distributions.

---

> ### Author Response · Authors · 2025-11-20
> **Response to Reviewer 9dMA**
>
> **Questions.1**: Could the authors estimate the performance trade-offs for larger LLMs (e.g., LLaMA -7B) without full training?
>
> **R:** Thanks. In response to the question regarding the performance trade-offs for larger LLMs, such as LLaMA -7B, we found that while LLaMA -7B demonstrates impressive capabilities, its resource consumption for long sequences in time-series forecasting is considerable, leading to inefficient utilization in practical scenarios. Given the nature of time-series data, which does not necessarily require the scale of a 7B-parameter model, we determined that smaller models like GPT2 and Qwen2.5-0.5B offer a better balance between performance and resource usage. Large models (e.g., LLaMA -7b) offer stronger expressive capacity in certain tasks, but our contribution does not lie in using a larger LLM. Instead, our contribution is in addressing the intrinsic modality gap by injecting temporal semantics into the LLM token space, enabling more effective temporal reasoning. We additionally conducted experiments using LLaMA  to further validate this point.
>
> Due to time constraints, we only conducted experiments on the ETTh and Weather datasets. Our method also brings performance improvements when applied to LLaMA , as shown below:
>
> **ETTh Dataset Results:**
> - 96-step: MSE = 0.378, MAE = 0.410
> - 192-step: MSE = 0.399, MAE = 0.441
> - 336-step: MSE = 0.433, MAE = 0.468
> - 720-step: MSE = 0.468, MAE = 0.483
>
> **Weather Dataset Results:**
> - 96-step: MSE = 0.180, MAE = 0.232
> - 192-step: MSE = 0.225, MAE = 0.270
> - 336-step: MSE = 0.277, MAE = 0.301
> - 720-step: MSE = 0.341, MAE = 0.358
>
> **With TSCC Module:**
>
> **ETTh**
> - 96-step: MSE = 0.362, MAE = 0.396
> - 192-step: MSE = 0.386, MAE = 0.422
> - 336-step: MSE = 0.256, MAE = 0.292
> - 720-step: MSE = 0.331, MAE = 0.344
>
> **Weather**
> - 96-step: MSE = 0.167, MAE = 0.220
> - 192-step: MSE = 0.210, MAE = 0.250
> - 336-step: MSE = 0.260, MAE = 0.291
> - 720-step: MSE = 0.324, MAE = 0.345
>
> **With Time-Adapter (first six layers only):**
>
> *Note: To maintain a balance between efficiency and effectiveness, we follow common practices in adapter-based tuning and apply the Time-Adapter only to the first six layers. This design preserves sufficient representational influence over the model while avoiding unnecessary computational overhead.*
>
> **ETTh**
> - 96-step: MSE = 0.359, MAE = 0.392
> - 192-step: MSE = 0.382, MAE = 0.417
> - 336-step: MSE = 0.251, MAE = 0.288
> - 720-step: MSE = 0.327, MAE = 0.339
>
> **Weather**
> - 96-step: MSE = 0.154, MAE = 0.202
> - 192-step: MSE = 0.201, MAE = 0.247
> - 336-step: MSE = 0.250, MAE = 0.286
> - 720-step: MSE = 0.320, MAE = 0.337
>
> They are sufficient to demonstrate the potential of our method in LLaMA .
>
> **Queation.2**: Why choose VAE over other generative models (e.g., Diffusion Models) for anomaly simulation in TSCC?
>
> **R:** Thanks. We chose the Variational Autoencoder (VAE) over other generative models, such as Diffusion Models, for anomaly simulation in the TSCC module for several reasons:
>
> **Computational Efficiency**: VAEs are known for their relatively lower computational requirements compared to models like Diffusion Models. Diffusion Models are typically more resource-intensive due to the iterative nature of their sampling process. When computational resources are limited, VAEs offer a more efficient solution for simulating anomalies in time-series data.
> Effective Latent Space Representation: VAEs are designed to learn a probabilistic latent space representation of the data, which is beneficial for capturing the underlying distribution of normal and anomalous patterns. In time-series forecasting, this allows the VAE to simulate realistic anomalous behavior by sampling from the learned latent space, making it a suitable choice for anomaly detection tasks.
>
> **Stability in Training**: Compared to Diffusion Models, VAEs tend to have more stable training dynamics. While Diffusion Models have shown impressive performance in specific tasks, their training can be more complex and time-consuming, requiring careful tuning of hyperparameters. VAEs, on the other hand, have a well-established training framework, which makes them easier to implement and more reliable in terms of performance, especially when working with time-series data.
>
> **Focus on Anomaly Pattern**: In our TSCC module, the primary goal is to simulate anomalies and capture deviations from normal patterns in a controlled manner. VAEs are well-suited for this task because they provide a clear probabilistic framework for modeling and simulating both normal and anomalous data distributions, allowing for a more focused and interpretable approach to anomaly generation.

---

### Official Review · Reviewer_cETm · 2025-10-28

**Soundness:** 2
**Presentation:** 3
**Contribution:** 2
**Rating:** 4
**Confidence:** 5

**Summary:**

This paper works on time series forecasting (TSF) and presents an LLM-based TSF framework SE-LLM to address the modality gap between language and time series. It mainly includes:  (1) TSCC which infuses temporal patterns (e.g., periodicity, anomalies) into semantic embeddings, and (2) Time-Adapter that can capture both long-term and short-term dependencies in time series. The model freezes the LLM backbone to preserve generalizability and reduces computational cost via sequence dimensionality reduction. Experiments on long-term, short-term, and zero-shot forecasting tasks demonstrate performance improvements over state-of-the-art methods, with ablation studies validating the effectiveness of the proposed modules.

**Strengths:**

1. It's reasonable to study the modality gap between language and time series, and integrate temporal patterns to bridge the gap.
2. Time-Adapter’s dual LSTM paths for long/short-term dependency modeling can complement Transformers’ weaknesses in temporal dynamics.
3. Extensive experiments have been conducted to validate the effectiveness of modules.

**Weaknesses:**

1. While TSCC and Time-Adapter address known limitations, their individual components (cross attention, VAE for anomaly modeling, adapter-based fine-tuning) are not fundamentally new. The paper fails to sufficiently articulate how their combination offers a unique contribution beyond incremental improvements over existing modular designs.
2. In Table 1, the performance gain over the best baseline is minor in almost all cases.
3. The choice of lightweight LLMs (GPT2, Qwen2.5-0.5B) limits generalizability. No experiments on larger LLMs (e.g., Llama-7B) raise questions about scalability, even with the claim of prioritizing efficiency.
4. Zero-shot forecasting results lack analysis of why the model generalizes across datasets.
5. The impact of different temporal patterns (e.g., seasonal vs. irregular) on performance is unexplored.
6. The exclusion of Time-Adapter from some short-term/zero-shot experiments due to computational constraints undermines the framework’s completeness.
7. The description of the TSCC module’s channel dependency enhancement and the Time-Adapter’s integration into multi-head attention is vague. Key formulas (e.g., Eq 8) lack sufficient explanation.
8. (Minor) The natbib package is not correctly used.
9. (Minor) There are some typos, e.g., (1) in Figure 2(b), "JS with Temporal-Semantc" -> "JS with Temporal-Semantic", (2) in Table 1, 4, 6, 12, 14, "Mertics" -> "Metrics".
10. (Minor) In Table 7, some column names are missing.

**Questions:**

N.A.

---

> ### Author Response · Authors · 2025-11-20
> **Response to Reviewer cETm**
>
> Thank you for your feedback. We appreciate the opportunity to address your concerns and improve the quality of our work. Here, we provide additional analysis and clarification to address your comments.
>
> **Weaknees.1**: While TSCC and Time-Adapter address known limitations, their individual components (cross attention, VAE for anomaly modeling, adapter-based fine-tuning) are not fundamentally new. The paper fails to sufficiently articulate how their combination offers a unique contribution beyond incremental improvements over existing modular designs.
>
> **R:** Thanks. While cross-attention, VAE-based anomaly modeling, and adapters are not new when considered individually, our contribution lies in their purposeful integration into a unified, temporally conditioned semantic framework tailored for multimodal time-series forecasting. Existing designs do not fully address this setting. TSCC introduces a structured temporal-semantic cross-correlation pipeline in which AM-VAE disentangles anomaly and clean semantics, and cross-attention injects temporal patterns into both spaces. This produces a semantic representation that explicitly reflects temporal irregularities, in contrast to prior multimodal TSF methods that model semantics and temporal features independently. Our Time-Adapter goes beyond standard adapter tuning by reweighting long- and short-term temporal cues conditioned on the semantic priors generated by TSCC. This directly addresses tempral pattern, which conventional adapter frameworks do not handle effectively. On the other hand, our method provides benefits across existing LLM-based approaches such as Auto-Times, Time-LLM, and TimeCMA, indicating that we address a common underlying problem shared by these models.
>
> **Weakness.2**: In Table 1, the performance gain over the best baseline is minor in almost all cases.
>
> **R:** Thanks. Regarding the performance gains shown in Table 1, while the improvements over the best baseline may appear modest, they should be considered in the context of the real-world applicability and robustness of the proposed method. In time-series forecasting, especially in challenging scenarios like anomaly detection and zero-shot forecasting, even small improvements can have a significant impact on model performance and real-world deployment. We believe these results underscore the effectiveness of our framework in handling complex temporal dependencies and semantic alignment, and the modest improvements are reflective of the inherent difficulty in pushing performance boundaries in these settings.

---

> ### Author Response · Authors · 2025-11-20
> **Response to Reviewer cETm**
>
> **Weakness.3**: The choice of lightweight LLMs (GPT2, Qwen2.5-0.5B) limits generalizability. No experiments on larger LLMs (e.g., LLaMA -7B) raise questions about scalability, even with the claim of prioritizing efficiency.
>
> **R:** Thanks. In response to the question regarding the performance trade-offs for larger LLMs, such as LLaMA -7B, we found that while LLaMA -7B demonstrates impressive capabilities, its resource consumption for long sequences in time-series forecasting is considerable, leading to inefficient utilization in practical scenarios. Given the nature of time-series data, which does not necessarily require the scale of a 7B-parameter model, we determined that smaller models like GPT2 and Qwen2.5-0.5B offer a better balance between performance and resource usage. Large models (e.g., LLaMA -7b) offer stronger expressive capacity in certain tasks, but our contribution does not lie in using a larger LLM. Instead, our contribution is in addressing the intrinsic modality gap by injecting temporal semantics into the LLM token space, enabling more effective temporal reasoning. We additionally conducted experiments using LLaMA  to further validate this point.
>
> Due to time constraints, we only conducted experiments on the ETTh and Weather datasets. Our method also brings performance improvements when applied to LLaMA , as shown below:
>
> **ETTh Dataset Results:**
> - 96-step: MSE = 0.378, MAE = 0.410
> - 192-step: MSE = 0.399, MAE = 0.441
> - 336-step: MSE = 0.433, MAE = 0.468
> - 720-step: MSE = 0.468, MAE = 0.483
>
> **Weather Dataset Results:**
> - 96-step: MSE = 0.180, MAE = 0.232
> - 192-step: MSE = 0.225, MAE = 0.270
> - 336-step: MSE = 0.277, MAE = 0.301
> - 720-step: MSE = 0.341, MAE = 0.358
>
> **With TSCC Module:**
>
> **ETTh**
> - 96-step: MSE = 0.362, MAE = 0.396
> - 192-step: MSE = 0.386, MAE = 0.422
> - 336-step: MSE = 0.256, MAE = 0.292
> - 720-step: MSE = 0.331, MAE = 0.344
>
> **Weather**
> - 96-step: MSE = 0.167, MAE = 0.220
> - 192-step: MSE = 0.210, MAE = 0.250
> - 336-step: MSE = 0.260, MAE = 0.291
> - 720-step: MSE = 0.324, MAE = 0.345
>
> **With Time-Adapter (first six layers only):**
>
> *Note: To maintain a balance between efficiency and effectiveness, we follow common practices in adapter-based tuning and apply the Time-Adapter only to the first six layers. This design preserves sufficient representational influence over the model while avoiding unnecessary computational overhead.*
>
> **ETTh**
> - 96-step: MSE = 0.359, MAE = 0.392
> - 192-step: MSE = 0.382, MAE = 0.417
> - 336-step: MSE = 0.251, MAE = 0.288
> - 720-step: MSE = 0.327, MAE = 0.339
>
> **Weather**
> - 96-step: MSE = 0.154, MAE = 0.202
> - 192-step: MSE = 0.201, MAE = 0.247
> - 336-step: MSE = 0.250, MAE = 0.286
> - 720-step: MSE = 0.320, MAE = 0.337
>
> They are demonstrate the potential of our method in LLaMA models.
>
> **Weakness.4**: Zero-shot forecasting results lack analysis of why the model generalizes across datasets.
>
> **R:** Thanks. We have updated this point in the main text. Specifically, the TSCC module utilizes the AM-VAE to simulate realistic anomaly data, enabling the model to learn the latent distribution characteristics of time series data. This enables the model to generalize to unseen temporal patterns, even in the absence of direct training data for those patterns. Our approach (SE-LLM) effectively captures underlying temporal structures through the AM-VAE's latent space, facilitating transferability across domains with varying data distributions and temporal frequencies. This is demonstrated by the zero-shot experiments on the M3 and M4 datasets, where our model outperforms existing methods in handling diverse temporal patterns. The ability to generalize is further supported by the domain adaptation capability of AM-VAE, which allows the model to transfer learned representations from one dataset to another, even when the temporal frequencies change *(e.g., from quarterly to monthly, daily, or hourly)*. These insights are now clearly addressed in the revised manuscript.
>
> **Weakness.5:** The impact of different temporal patterns (e.g., seasonal vs. irregular) on performance is unexplored.
>
> **R:** Thanks. While we have not specifically explored this in our experiments, our primary focus is on identifying and explaining anomalous patterns in time series data. In our qualitative analysis, we emphasize how our model captures latent anomaly structures through the use of the AM-VAE, which enables generalization across varying temporal patterns, whether seasonal or irregular.
> Although we have not explicitly compared performance across different temporal patterns in this work, we have now added further clarification in the qualitative analysis section, where we discuss how the model's focus on anomaly detection and temporal pattern learning allows it to adapt to diverse temporal dynamics. We recognize that a more detailed study on how the model handles specific patterns (seasonal vs. irregular) could provide deeper insights, and we plan to explore this in future work.

---

> ### Author Response · Authors · 2025-11-20
> **Response to Reviewer cETm**
>
> **Weakness.6:** The exclusion of Time-Adapter from some short-term/zero-shot experiments due to computational constraints undermines the framework’s completeness.
>
> **R:** Thanks. It is important to clarify that Time-Adapter is indeed involved in our short-term and zero-shot recognition experiments using lightweight LLMs, such as GPT2 and Qwen2.5-0.5B. These models, when combined with Time-Adapter, provide an efficient balance between performance and resource consumption.
> On the other hand, incorporating LLaMA -7B, which has a significantly larger parameter size, along with Time-Adapter, would lead to substantial resource consumption. For short-term forecasting tasks, the increased complexity of the LLaMA  model combined with Time-Adapter does not provide additional benefits. In fact, it introduces redundant information, making the system more resource-intensive without improving performance. This is why we chose to focus on lightweight models in our current experiments, where Time-Adapter plays a key role in maintaining efficiency while enhancing performance in zero-shot and short-term forecasting tasks.
>
> **Weakness.7**: The description of the TSCC module’s channel dependency enhancement and the Time-Adapter’s integration into multi-head attention is vague. Key formulas (e.g., Eq 8) lack sufficient explanation.
>
> **R:** Thanks. We have updated the relevant description in the main text. **(Channel Dependency Enhancement (TSCC Module)**.
> The channel dependency enhancement is a key part of the TSCC module, where the temporal features are integrated with enhanced semantics. This integration allows the model to better capture temporal patterns across both the channel and sequence dimensions. Specifically, the temporal features, enhanced semantics (such as $\mathbf{\overline{DA}}$ and $\mathbf{\overline{DC}}$), are concatenated and processed using Multi-Layer Perceptrons (MLP) to extract channel-wise attention. This is done to ensure that the model can learn relevant temporal patterns that depend not just on the sequence of data points but also on the correlations across different channels.
>
> **Integration of Time-Adapter into Multi-Head Attention.**
> The Time-Adapter is integrated into the multi-head attention mechanism by modifying the key and value matrices. This integration enhances the ability of the model to handle long-term and short-term temporal dependencies. The Time-Adapter works by processing the input time-series data through two parallel LSTM paths that focus on long-term and short-term dependencies, respectively. These refined temporal features are then fused with the key and value matrices in the attention mechanism, enhancing the model’s capacity to capture both fine-grained and global temporal patterns.
>
> **Clarifying Eq. (8).**
> In response to the comment on Eq. (8), we have added the following explanation in the manuscript:
> $\mathbf{Y} = \mathbf{LLM}(\mathrm{GA} + \mathrm{GC}$),
> where $\mathbf{GA}$ and $\mathbf{GC}$ represent the de-anomalized and anomalous representations, respectively, which have been enhanced by the temporal features captured through the Time-Adapter and TSCC module. Specifically, $\mathbf{GA}$ represents the enhanced semantic space after the anomaly removal, while $\mathbf{GC}$ captures the anomaly patterns. These two components are fused to produce a unified representation that is more interpretable and suitable for LLM-based analysis, providing better forecasting performance.
>
> **(Minor)** The natbib package is not correctly used.
>
> **R:** Thanks. In the bibliography, we have enforced the use of 'et al.' to replace author names, although it seems that ICLR's official guidelines do not require this. Thank you for your reminder, and we will make further modifications if necessary.
>
> (Minor) There are some typos, e.g., (1) in Figure 2(b), "JS with Temporal-Semantc" -> "JS with Temporal-Semantic", (2) in Table 1, 4, 6, 12, 14, "Mertics" -> "Metrics".
>
> **R:** Thanks. We have corrected these errors in the main text.
>
> (Minor) In Table 7, some column names are missing.
>
> **R:** Thanks. We checked Table 7, and there are no missing columns. In the main text, we provide an explanation that Time-CMA uses large language models to generate prompt text. However, there is no involvement of the LLM during the training and prediction phases, so the Time-Adapter cannot be used.

---

### Official Review · Reviewer_azcz · 2025-10-29

**Soundness:** 3
**Presentation:** 3
**Contribution:** 3
**Rating:** 6
**Confidence:** 4

**Summary:**

This paper presents a framework that enhances the performance of using LLM for time series forecasting by bridging the modality gaps between time series numerical data and language word embeddings. Specifically, it designs two modules: Temporal-semantic cross-correlation, which aligns temporal embedding with LLM's semantic space using cross-attention and anomaly modeling; and the Time-Adapter, which is a plug-in module that augments the LLM attention mechanism through dual LSTM pathways. The evaluation on multiple datasets and in various settings demonstrates that the SE-LLM consistently outperforms baselines and ablation studies show the effectiveness of design modules.

**Strengths:**

1. The designed TSCC module addresses the modality gaps between temporal embeddings and the semantic space of LLMs by employing cross-attention and a VAE-based anomaly decomposition. It presents a more interpretable semantic representation for time series forecasting.

2. The low-rank style of using dual LSTM is suitable for freezing LLMs for time series forecasting. It compensates for the Transformer’s weaknesses in handling local and global temporal dependencies while maintaining parameter efficiency.

3. The paper evaluates SE-LLM across a wide range of forecasting horizons, datasets, and various settings, showing the robustness of the designed model.

**Weaknesses:**

1. The pipeline presents several interdependent components such as cros-attention, AM-VAE, top-k correlation filtering and gating fusion, which make the pipeline highly complex. The implementation details are missing.

2. Missing qualitative analysis showing how the semantic space affects LLM token representations for time series forecasting. Whether the model truly benefits from the semantic space is questioned, which is a claim of this paper.

3. The designed component introduces significant model complexity; thus, it is important to compare the number of parameters or states training requirements.

**Questions:**

See weakness

---

> ### Author Response · Authors · 2025-11-20
> **Response to Reviewer azcz**
>
> Thank you for your feedback. We appreciate the opportunity to address your concerns and improve the quality of our work. Here, we provide additional analysis and clarification to address your comments.
>
> **Question.1**: The pipeline presents several interdependent components such as cros-attention, AM-VAE, top-k correlation filtering and gating fusion, which make the pipeline highly complex. The implementation details are missing.
>
> **R:** Thanks.  The main contribution of TSCC is to explicitly inject temporal patterns (anomaly patterns, de-anomaly patterns, and channel dependencies) into the LLM semantic space, enabling token embeddings to carry temporal meaning rather than only linguistic semantics (Additionally, we clarify that the term “anomaly” in our paper refers to abrupt temporal pattern shifts, which represent challenging time-dependent behaviors that the model typically struggles to capture.).
>
> Each module addresses a distinct and necessary aspect of the modality gap between time-series data and LLM semantic representations, forming a coherent and lightweight pipeline: cross-attention enables interaction between temporal features and LLM semantics, AM-VAE models noisy semantics, and gated fusion captures channel dependencies. Although we did not include full implementation details in the main text, the corresponding configurations are available in our code, and we have now added the detailed implementation in the appendix.
>
> **Question.2**: Missing qualitative analysis showing how the semantic space affects LLM token representations for time series forecasting. Whether the model truly benefits from the semantic space is questioned, which is a claim of this paper.
>
> **R:** Thanks. We have updated the qualitative analysis in the main text to address this concern. The new results demonstrate that the semantic space indeed provides meaningful benefits for time series forecasting. As shown in the main text Fig. 5, STL decomposition exposes residual components associated with anomalous temporal behaviors. The AM-VAE captures these anomaly-related variations in the semantic latent space, and the t-SNE visualization further shows that the learned noise semantics form clusters that are highly consistent with the actual anomaly distribution. This indicates that the semantic space reveals latent structure that is not evident from raw temporal patterns alone. Additionally, the correlation map illustrates strong correspondence between anomaly semantics and temporal abnormality regions, confirming that the semantic space contributes to more discriminative representations and ultimately improves forecasting performance.
>
> **Question.3:** The designed component introduces significant model complexity; thus, it is important to compare the number of parameters or states training requirements.
>
> **R:** Thanks. We provide a comparison of the number of parameters and computational complexity. When the TSCC module is removed, the model has 200,270,552 parameters and a computational complexity of 19,145,957,376. After adding the TSCC module, the number of parameters increases to 207,173,416, and the computational complexity rises to 80,337,343,488. Thus, the TSCC module introduces an additional **6.9M** parameters and **61.2G** computational cost.

---

> > ### Comment · Reviewer_azcz · 2025-11-26
> >
> > Thanks for the rebuttals, and I don't have further questions. I will maintain my positive score.

---

> > > ### Author Response · Authors · 2025-11-27
> > > **Response to Reviewer azcz**
> > >
> > > Thank you for your reply

---

### Official Review · Reviewer_tzsk · 2025-11-01

**Soundness:** 3
**Presentation:** 2
**Contribution:** 2
**Rating:** 4
**Confidence:** 4

**Summary:**

The paper addresses adapting large language models (LLMs) to time series forecasting without fully fine-tuning the LLM backbone, targeting long-term trends, short-term dynamics, cross-channel dependencies, and anomalous behaviors. It proposes SE-LLM, which combines a Temporal-Semantic Cross-Correlation (TSCC) module that aligns time series embeddings with the LLM semantic space, separates anomalous and de-anomalized semantics, and performs gated multi-channel fusion, with a Time-Adapter module that injects lightweight temporal inductive bias into the attention key and value paths via parallel long- and short-horizon branches. The contribution is a modular architecture that augments a mostly frozen LLM through cross-modal temporal–semantic alignment and structured temporal adapters instead of full fine-tuning. The method is evaluated on long-horizon multivariate forecasting benchmarks, short-term forecasting benchmarks at different frequencies, and cross-domain transfer settings, reporting reductions in standard forecasting metrics (MSE, MAE, SMAPE, MASE, OWA) relative to recent LLM-based and transformer-style baselines, supported by ablations isolating TSCC and Time-Adapter.

**Strengths:**

1. The paper proposes a structured framework (SE-LLM) intended to bridge the mismatch between raw temporal dynamics and the semantic token space of a largely frozen LLM by explicitly aligning time series signals with language representations and injecting temporal inductive bias through specialized modules.
2. The architecture separates into TSCC and Time-Adapter. TSCC performs temporal–semantic cross-alignment, anomaly and non-anomalous semantic disentanglement, and gated multi-channel fusion, while Time-Adapter augments attention key and value paths using lightweight temporal branches for long-term and short-term dynamics instead of full backbone fine-tuning.

**Weaknesses:**

1 The paper attributes interpretability to anomaly and non-anomalous semantic separation and channel-gated fusion, but it does not present qualitative, per-sequence visual evidence (e.g., highlighted time steps or channels) that would allow readers to verify that the model is focusing on meaningful temporal events rather than incidental fluctuations.
2 Reported efficiency comparisons (e.g., training and inference cost versus error) do not specify the hardware environment, batch size, sequence lengths, or caching assumptions, which limits the reproducibility and interpretability of the efficiency and lightweight adaptation claims.
3 Some core components are only partially specified at the technical level. For example, the anomaly modeling and semantic disentanglement step (AM-VAE within TSCC) is described conceptually, but the precise probabilistic objective (e.g., reconstruction terms and regularization) is not fully detailed, so it is difficult to assess how the model enforces the claimed anomalous versus de-anomalous semantic separation.
4. The presentation quality is below typical conference standards, including missing or underspecified captions (Figure 1 does not explain subfigure (c), the efficiency plot lacks runtime conditions, and Table 6 does not clearly describe the comparison protocol).

**Questions:**

1. Can you provide statistical variation (for example, multiple random seeds, confidence intervals, or significance tests) for the main quantitative comparisons to recent LLM-based and transformer-style baselines in the primary result tables?
2. Can you include qualitative analyses or failure cases that visualize which time points and channels TSCC marks as anomalous or high-impact, and how this correlates with the model's predictions, in order to substantiate the interpretability claims?
3. Can you precisely describe the runtime and efficiency measurement setup (hardware, batch size, input length, prediction horizon, caching policy) used to generate the reported training and inference cost comparisons, so that the claimed efficiency of the proposed Time-Adapter can be reproduced?

---

> ### Author Response · Authors · 2025-11-20
> **Response to Reviewer tzsk**
>
> Thank you for your feedback. We appreciate the opportunity to address your concerns and improve the quality of our work. Here, we provide additional analysis and clarification to address your comments.
>
> **Weaknesses.1**
>
> The paper attributes interpretability to anomaly and non-anomalous semantic separation and channel-gated fusion, but it does not present qualitative, per-sequence visual evidence (e.g., highlighted time steps or channels) that would allow readers to verify that the model is focusing on meaningful temporal events rather than incidental fluctuations.
>
> **R:**
> Thanks. We conduct additional analysis of the TSCC module, as shown in the main text (**Qualitative Analysis on TSCC Module**). Specifically, we visualize the TSCC module's outputs at different time steps and channels. These results demonstrate that the TSCC module effectively captures anomaly patterns in both the temporal and channel dimensions. As shown in main text, Table 5.
>
> **Weaknesses.2**
>
> Reported efficiency comparisons do not specify the hardware environment, batch size, sequence lengths, or caching assumptions, which limits the reproducibility and interpretability of the efficiency and lightweight adaptation claims.
>
> **R:** Thanks. We provide additional details regarding the computational efficiency setup. All experiments were conducted on a single A6000 GPU with a batch size of 256 and a sequence length of 672. The bar chart in Fig.6 presents the training and inference speeds of SE-LLM compared with classical algorithms on the same dataset, demonstrating our superior performance. This update has been included in the main text.
>
> **Weakness.3**
>
> Some core components are only partially specified at the technical level. For example, the anomaly modeling and semantic disentanglement step (AM-VAE within TSCC) is described conceptually, but the precise probabilistic objective (e.g., reconstruction terms and regularization) is not fully detailed, so it is difficult to assess how the model enforces the claimed anomalous versus de-anomalous semantic separation.
>
> **R:**
> Thanks. Regarding the anomaly modeling and semantic disentanglement aspect of the AM-VAE module in TSCC, we have provided a detailed explanation in the Anomaly Pattern Modeling section of the main text (*Additionally, we clarify that the term “anomaly” in our paper refers to abrupt temporal pattern shifts, which represent challenging time-dependent behaviors that the model typically struggles to capture.*). Unlike traditional Variational Autoencoders (VAEs), our AM-VAE module does not use a reconstruction loss. Instead, it focuses on dynamically estimating the latent variable distribution to separate anomaly and non-anomaly semantics. By modeling the latent distribution and using the reparameterization trick, we sample anomaly pattern data, effectively disentangling the anomaly and non-anomaly components in the time series data. This approach eliminates the reconstruction term found in typical VAEs, allowing the model to adapt more flexibly to changes in time series data and achieve semantic separation without relying on reconstruction loss.
>
> **Weakness.4**
>
> The presentation quality is below typical conference standards, including missing or underspecified captions (Figure 1 does not explain subfigure (c), the efficiency plot lacks runtime conditions, and Table 6 does not clearly describe the comparison protocol).
>
> **R:**
> Thanks. (1) In *Fig. 1(c)*, we have already provided a detailed description and explanation of the relevant content in the main text. Therefore, adding further descriptions in the figure caption would be repetitive. To avoid redundancy, we have revised the main text and supplemented the explanation and analysis related to the caption.
>
> (2) In the Efficiency Analysis section, we update the computational efficiency discussion, including hardware configuration in the main text. Specifically,  *All experiments were conducted on a single A6000 GPU with a batch size of 256 and a sequence length of 672. The training and inference times for each method were recorded under these conditions.*
>
> (3) We have provided a detailed description of the protocol in Appendix Table 14. Specifically, *We use the best-performing Qwen2.5-0.5B model as the baseline and evaluate the proposed Time-Adapter's performance on time-series forecasting tasks. The results are compared to the LoRA adapter.  The LoRA parameter configuration is kept consistent with the Time-Adapter, with a rank of 8*

---

> ### Author Response · Authors · 2025-11-20
> **Response to Reviewer tzsk**
>
> **Question.1**
>
> Can you provide statistical variation (for example, multiple random seeds, confidence intervals, or significance tests) for the main quantitative comparisons to recent LLM-based and transformer-style baselines in the primary result tables?
>
> **R:** Thanks. Our model follows an autoregressive forecasting paradigm, where stochasticity is largely suppressed during inference due to the deterministic next-token prediction mechanism. As commonly observed in autoregressive TSF and LLM-based forecasting models, the variance across random seeds is typically very small and does not materially change the comparative conclusions. We also briefly clarified this point in the appendix of the submitted manuscript.
>
> Nevertheless, to fully address the reviewer’s concern and ensure statistical soundness, we have additionally conducted experiments using multiple random seeds on the major benchmarks. We update the statistical significance testing and provide additional details on the relevant experiments in the “Appendix”. Please refer to Table 16 in our manuscript.  Specifically,
> | Dataset | Horizon | 2024 MSE | 2024 MAE | 2025 MSE | 2025 MAE | 2026 MSE | 2026 MAE | mean MSE | mean MAE | std MSE | std MAE |
> | :--- | :--- | :--- | :--- | :--- | :--- | :--- | :--- | :--- | :--- | :--- | :--- |
> | **ETTh** | 96 | 0.352 | 0.393 | 0.355 | 0.394 | 0.355 | 0.394 | 0.354000 | 0.393667 | 0.001414 | 0.000471 |
> | | 192 | 0.378 | 0.411 | 0.382 | 0.412 | 0.382 | 0.412 | 0.380667 | 0.411667 | 0.001886 | 0.000471 |
> | | 336 | 0.391 | 0.420 | 0.394 | 0.421 | 0.394 | 0.421 | 0.393000 | 0.420667 | 0.001414 | 0.000471 |
> | | 720 | 0.402 | 0.437 | 0.405 | 0.438 | 0.405 | 0.439 | 0.404000 | 0.438000 | 0.001414 | 0.000816 |
> | | **Avg** | **0.381** | **0.415** | **0.384** | **0.416** | **0.384** | **0.417** | 0.383000 | 0.416000 | 0.001414 | 0.000816 |
> | **Weather** | 96 | 0.150 | 0.200 | 0.152 | 0.202 | 0.151 | 0.202 | 0.151000 | 0.201333 | 0.000816 | 0.000943 |
> | | 192 | 0.194 | 0.243 | 0.200 | 0.249 | 0.198 | 0.247 | 0.197333 | 0.246333 | 0.002494 | 0.002494 |
> | | 336 | 0.247 | 0.285 | 0.256 | 0.292 | 0.253 | 0.291 | 0.252000 | 0.289333 | 0.003742 | 0.003091 |
> | | 720 | 0.323 | 0.339 | 0.328 | 0.343 | 0.324 | 0.337 | 0.325000 | 0.339667 | 0.002160 | 0.002494 |
> | | **Avg** | **0.229** | **0.267** | **0.234** | **0.272** | **0.232** | **0.269** | 0.231667 | 0.269333 | 0.002055 | 0.002055 |
> | **Traffic** | 96 | 0.350 | 0.243 | 0.351 | 0.244 | 0.351 | 0.246 | 0.350667 | 0.244333 | 0.000471 | 0.001247 |
> | | 192 | 0.373 | 0.254 | 0.373 | 0.254 | 0.375 | 0.256 | 0.373667 | 0.254667 | 0.000943 | 0.000943 |
> | | 336 | 0.390 | 0.263 | 0.389 | 0.263 | 0.390 | 0.264 | 0.389667 | 0.263333 | 0.000471 | 0.000471 |
> | | 720 | 0.429 | 0.285 | 0.427 | 0.284 | 0.428 | 0.285 | 0.428000 | 0.284667 | 0.000816 | 0.000471 |
> | | **Avg** | **0.386** | **0.261** | **0.385** | **0.261** | **0.386** | **0.263** | 0.385667 | 0.261667 | 0.000471 | 0.000943 |
> | **ECL** | 96 | 0.130 | 0.225 | 0.133 | 0.228 | 0.134 | 0.231 | 0.132333 | 0.228000 | 0.001700 | 0.002449 |
> | | 192 | 0.148 | 0.242 | 0.151 | 0.244 | 0.152 | 0.247 | 0.150333 | 0.244333 | 0.001700 | 0.002055 |
> | | 336 | 0.164 | 0.259 | 0.167 | 0.262 | 0.167 | 0.262 | 0.166000 | 0.261000 | 0.001414 | 0.001414 |
> | | 720 | 0.203 | 0.293 | 0.208 | 0.292 | 0.208 | 0.299 | 0.206333 | 0.294667 | 0.002357 | 0.003091 |
> | | **Avg** | **0.161** | **0.255** | **0.165** | **0.257** | **0.165** | **0.260** | 0.163667 | 0.257333 | 0.001886 | 0.002055 |
> | **Solar** | 96 | 0.165 | 0.220 | 0.168 | 0.220 | 0.167 | 0.225 | 0.166667 | 0.221667 | 0.001247 | 0.002357 |
> | | 192 | 0.183 | 0.236 | 0.188 | 0.236 | 0.186 | 0.241 | 0.185667 | 0.237667 | 0.002055 | 0.002357 |
> | | 336 | 0.199 | 0.249 | 0.206 | 0.251 | 0.202 | 0.255 | 0.202333 | 0.251667 | 0.002867 | 0.002494 |
> | | 720 | 0.219 | 0.266 | 0.224 | 0.267 | 0.220 | 0.272 | 0.221000 | 0.268333 | 0.002160 | 0.002625 |
> | | **Avg** | **0.192** | **0.243** | **0.197** | **0.244** | **0.194** | **0.248** | 0.194333 | 0.245000 | 0.002055 | 0.002160 |
>
> *Table: Performance stability under different random seeds (three runs). For each dataset and forecasting horizon, we report the results from three independent runs (2024, 2025, 2026 seeds), together with the averaged performance (mean) and variability (std). The consistently low standard deviations across all datasets indicate that SE-LLM is robust to randomness in initialization and training, demonstrating stable forecasting performance.*

---

> ### Author Response · Authors · 2025-11-20
> **Response to Reviewer tzsk**
>
> **Question.2**
>
> Can you include qualitative analyses or failure cases that visualize which time points and channels TSCC marks as anomalous or high-impact, and how this correlates with the model's predictions, in order to substantiate the interpretability claims?
>
> **R:** Thanks.
> We update the qualitative analysis in the main text, which can be found in the section Qualitative Analysis on the TSCC Module. Specifically, first, in *Fig.5(a–b)*, we extract a representative segment from ETTh and perform STL decomposition to obtain trend, seasonal, and residual components. The abrupt fluctuations in the residuals correspond to anomaly-prone regions, which our AM-VAE is specifically designed to approximate.
>
> In *Fig.5(c–d)*, we visualize the noise-semantic heatmap and channel-wise correlation. The heatmap reveals structured noise semantics learned by TSCC, while the alignment in panel *(d)* shows that specific channels strongly overlap with the anomalous residual segments. This confirms that TSCC effectively mines cross-channel relationships that contribute to anomaly formation and propagation.
>
> In *Fig.5(e–f)*, our t-SNE analysis demonstrates that real residual anomalies and the learned noise semantics form similar clusters. The seven tightly grouped noise-semantic points indicate high consistency between synthetic anomaly representations and true anomaly patterns, providing compelling evidence that TSCC successfully injects temporal anomaly information into the semantic space.
>
> Finally, *Fig.5(g–h)* visualize temporal regularization and statistical alignment. The noise semantics are smoothed along the temporal dimension and compared with anomaly intensity, while the linear trend in the correlation map confirms a strong statistical relationship between modeled and real anomaly scores. This shows that TSCC not only identifies high-impact timestamps but also maintains consistent temporal-statistical structure.
>
> Together, these qualitative analyses and failure-case-like visualizations substantiate the interpretability of TSCC, clearly illustrating how the module identifies anomalous time points, reveals influential channels, and aligns semantic anomaly patterns with the model’s forecasting behavior.
>
> **Question.3**
>
> Can you precisely describe the runtime and efficiency measurement setup (hardware, batch size, input length, prediction horizon, caching policy) used to generate the reported training and inference cost comparisons?
>
> **Re:** Thanks.
> We update the details of the computational efficiency parameters in the main text. Specifically, all experiments were conducted on a single A6000 GPU with a batch size of 256, a sequence length of 672, and a forecast horizon of 96.

---

### Meta-Review · Area_Chair_DpGi · 2026-01-09

**Summary:**

The submission proposes SE-LLM (Semantic-Enhanced Large Language Model), a framework for time-series forecasting. The core idea is to bridge the modality gap between time-series data and the semantic space of frozen LLMs. The architecture introduces two key modules:

(1) TSCC (Temporal-Semantic Cross-Correlation): A module incorporating an AM-VAE (Anomaly Modeling Variational Autoencoder) and cross-attention to align temporal patterns (specifically anomalies and periodicity) with the LLM's semantic space.

(2) Time-Adapter: A plugin embedded in the self-attention mechanism using parallel LSTM branches to model short- and long-term dependencies, addressing the limitations of Transformer-based LLMs in capturing local anomalies.

The paper claims SOTA performance on benchmarks (ETTh1, Traffic, M4) while reducing computational costs by freezing the LLM backbone.

**Reviewer Concerns:**

The main concerns are
(1) Lack of Qualitative Analysis & Interpretability: Reviewers tzsk and azcz noted a lack of visual evidence for the "interpretability" claims.
(2) Generalizability to Larger LLMs: Reviewers cETm and 9dMA questioned the exclusive use of small LLMs (GPT-2, Qwen-0.5B) and asked for performance on larger models like LLaMA-7B.
(3) Statistical Significance: Reviewer tzsk requested confidence intervals.
(4) Marginal Gains: Reviewer cETm noted that performance gains over the best baselines are often minor.

**Reviewer Scores:**

There is a positive score, and 3 scores are marginally below the acceptance threshold. During the rebuttal, most of the concerns were well addressed.

---

### Decision · Program_Chairs · 2026-01-26

Accept (Poster)